# The Implicit Bias of Heterogeneity towards Invariance: A Study of Multi-Environment Matrix Sensing

**Yang Xu**[*]
School of Mathematical Sciences
Peking University
xuyang1014@pku.edu.cn

**Yihong Gu**[*]
Department of Operations Research and Financial Engineering
Princeton University
yihongg@princeton.edu

**Cong Fang**[†]
School of Intelligence Science and Technology
Peking University
fangcong@pku.edu.cn

## Abstract

Models are expected to engage in invariance learning, which involves distinguishing the core relations that remain consistent across varying environments to ensure the predictions are safe, robust and fair. While existing works consider specific algorithms to realize invariance learning, we show that model has the potential to learn invariance through standard training procedures. In other words, this paper studies the implicit bias of Stochastic Gradient Descent (SGD) over heterogeneous data and shows that the implicit bias drives the model learning towards an invariant solution. We call the phenomenon the *implicit invariance learning*. Specifically, we theoretically investigate the multi-environment low-rank matrix sensing problem where in each environment, the signal comprises (i) a lower-rank invariant part shared across all environments; and (ii) a significantly varying environment-dependent spurious component. The key insight is, through simply employing the large step size large-batch SGD sequentially in each environment without any explicit regularization, the oscillation caused by heterogeneity can provably prevent model learning spurious signals. The model reaches the invariant solution after certain iterations. In contrast, model learned using pooled SGD over all data would simultaneously learn both the invariant and spurious signals. Overall, we unveil another implicit bias that is a result of the symbiosis between the heterogeneity of data and modern algorithms, which is, to the best of our knowledge, first in the literature.

## 1 Introduction

In real applications, the machine learning models are often heavily over-parameterized, which means that the number of parameters exceeds the number of data. For over-parameterized models,

---

[*]Equal Contribution.
[†]Corresponding author.

38th Conference on Neural Information Processing Systems (NeurIPS 2024).

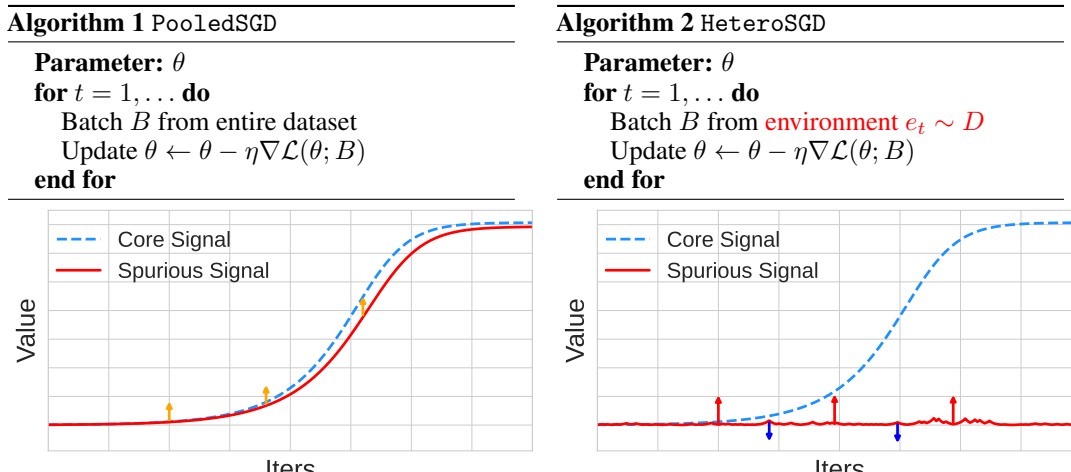

Figure 1: An illustration comparing training from aggregated data versus from heterogeneous data. The left example resembles the case where the model is trained on complete datasets, resulting in a stable spurious signal that the model tends to fit. The right example simulates a two-environment case where the spurious signal changes at each step. This oscillation creates a contraction effect, preventing the model from fitting the spurious signal.

the generalization in the general case becomes ill-posed. One key insight to generalize well is the implicit preference of the optimization algorithm which plays the role of regularization/bias [45, 22]. Nowadays, there are several kinds of implicit bias discovered from optimization algorithms under different models and settings. One common feature of the bias is the simplicity which concludes that (stochastic) gradient-based algorithms perform the incremental learning with the model complexity gradually increasing. Therefore, benign generalization is possible even when the number of training data is limited. For example, Li et al. [31], Gunasekar et al. [18] show that unregularized gradient descent can find the low-rank solution efficiently for matrix sensing models. Kalimeris et al. [27], Gissin et al. [16], Jiang et al. [24], and Jin et al. [25] further show that (Stochastic) Gradient Descent ((S)GD) learn models from simple ones to complex ones. Most of the existing works study the implicit bias of algorithms over a single distributional environment data.

However, data in modern practice are often collected from multiple sources, thus exhibiting certain heterogeneity. For example, medical data may come from multiple hospitals, and training sets for large language models consist of numerous corpus from the Internet [1]. So *what is the impact of implicit bias for standard training algorithms over heterogeneous data?*

This paper initializes the study and shows that implicit bias of SGD on an over-parameterized model using multi-environment heterogeneous data and shows that the implicit bias can not only save the number of training data but also, more importantly, drive the model learning the invariant relation across diverse environments.

Learning the invariant relation that remains consistent across varying environments [43] has garnered significant attention in recent years. Though the association-based standard machine learning pipelines can achieve a good performance with identical data distributions, a higher requirement is to make predictions robustly generalize over diverse downstream environments. Learning invariance produces reliable, fair, robust predictions against strong structural mechanism perturbation. More importantly, it opens the door to pursue causality blind to any prior knowledge and can unveil direct causes when the heterogeneity among environments is sufficient [17, 43]. While existing works consider specific algorithms to realize invariance learning, this work shows that implicit bias of algorithms over heterogeneous data has the potential to automatically learn the invaraince. We call the phenomenon *the implicit invariance learning*, partially explains why active invariance learning may not be necessary in practice [42]. Our key insight is:

*The heterogeneity of the data, and the large step size adopted in the optimization algorithm jointly provide strong multiplicative oscillations in the spurious signal space, which prevents the model from moving in the direction of unstable and spurious solutions, thus resulting in an implicit bias to the invariant solution.*

We illustrate it rigorously through a simple, canonical but insightful model – multi-environment matrix sensing, where in each environment the signal consists of two parts: an invariant low-rank matrix $\mathbf{A}^\star \in \mathbb{R}^{d \times d}$ and an environment-varying spurious low-rank matrix $\mathbf{A}^{(e)} \in \mathbb{R}^{d \times d}$ where environment $e \in \mathcal{E}$, the set of environments. For each environment $e \in \mathcal{E}$, the joint distribution of $(\mathbf{X}^{(e)}, y^{(e)})$ satisfies $y^{(e)} = \langle \mathbf{X}^{(e)}, \mathbf{A}^\star \rangle + \langle \mathbf{X}^{(e)}, \mathbf{A}^{(e)} \rangle$ with matrix inner product $\langle \mathbf{A}, \mathbf{B} \rangle = \text{Trace}(\mathbf{B}^\top \mathbf{A})$. Here $\mathbf{X}^{(e)} \in \mathbb{R}^{d \times d}$ is a random linear measurement and $y^{(e)} \in \mathbb{R}$ is the response. We consider the case that association does not coincide invariance (or causality), where averaging over all the environments, the best prediction of $y$ given $\mathbf{X}$ is

$$f^\star(\mathbf{X}) = \underbrace{\langle \mathbf{X}, \mathbf{A}^\star \rangle}_{invariant\ part} + \underbrace{\langle \mathbf{X}, \mathbb{E}_e[\mathbf{A}^{(e)}] \rangle}_{spurious\ part} \qquad \text{with} \qquad \mathbb{E}_e[\mathbf{A}^{(e)}] \neq 0.$$

In this case, it is not surprising that given enough data, the standard empirical risk minimizer algorithm, for example, running SGD on pooled data, will return a solution that converges to $f^\star$, which diverges from the invariant solution. In this paper, we will show that surprisingly, if each batch is sampled from data in one environment rather than data in all the environments, the heterogeneity in the environments together with the implicit regularization effects in the SGD algorithm can drive it towards the invariant solution. This can be stated informally as follows.

**Theorem 1** (Main result, informal). *Under a sufficient heterogeneity condition and some regularity conditions in matrix sensing, if we adopt an over-parameterized model and runs stochastic gradient descent where batches are sampled from one environment, i.e.,* `HeteroSGD` *(Algorithm 3), then*

$$\|\hat{\theta}_{\texttt{HeteroSGD}} - \mathbf{A}^\star\|_F = o_\mathbb{P}(1).$$

*Instead, the standard approach, i.e.,* `PooledSGD`*, will return solution $\hat{\theta}_{\texttt{PooledSGD}}$ satisfying*

$$\|\hat{\theta}_{\texttt{PooledSGD}} - \mathbf{A}^\star - \mathbf{A}^s\|_F = o_\mathbb{P}(1) \qquad thus \qquad \|\hat{\theta}_{\texttt{PooledSGD}} - \mathbf{A}^\star\|_F = \Omega_\mathbb{P}(1).$$

An illustration of our result is shown in Figure 1. Our result demonstrates that implicit bias of commonly used algorithms over heterogeneous data has the potential to drive the model to learn the invariant relation. Such a result thereby provides an explanation for why models may attain some robust and even causal prediction after SGD training.

We emphasize that the previous implicit bias studies are restricted to the same data distribution generalization, under which the population-level minimizer $f^\star$ minimizing the loss with infinite data is the target in pursuit. However, both the population-level minimizer and those "good" solutions under previous studies diverge from an invariant solution in general and are no longer benign in this context, this is termed as "curse of endogeneity" [12, 13].

**Notations.** We use the conventional notations $O(\cdot), o(\cdot), \Omega(\cdot)$ to ignore the absolute constants, $\tilde{O}(\cdot), \tilde{o}(\cdot), \tilde{\Omega}(\cdot)$ to further ignore the polynomial logarithmic factors. Similarly, $a \lesssim b$ means that there exists an absolute constant $C > 0$ such that $a \lesssim Cb$. We also denote it as $a \ll b$, $b \gg a$ if $a = o(b)$. Unless otherwise specified, we use lowercase bold letters such as $\mathbf{v}$ to represent vectors, and use $\|\mathbf{v}\|$ to denote its Euclidean norm. We use uppercase bold letters such as $\mathbf{X}$ to represent matrices and use $\|\mathbf{X}\|, \|\mathbf{X}\|_F, \|\mathbf{X}\|_*$ to denote its operator norm, Frobenius norm and nuclear norm, respectively. We use $\kappa(\mathbf{X})$ to denote the condition number, which is $\sigma_{\max}(\mathbf{X})/\sigma_{\min}(\mathbf{X})$. We define $Z = o_\mathbb{P}(1)$ if the random variable $Z$ satisfies $Z \xrightarrow{P} 0$.

## 2 Related Works

**Implicit Regularization.** It is believed that implicit bias is a key factor in why over-parameterized models can generalize well. Through the analysis of certain settings, existing results suggest that GD/SGD prefers solutions with specific properties [45, 19, 41, 38, 23], or specific local landscapes [3, 9, 32, 38]. For the matrix sensing problem, several works [18, 31, 27, 16, 46, 52, 24, 25] analyze

the (S)GD dynamics to show how (S)GD recovers the ground truth low-rank matrix. Recently, the effects of large step size have aroused much attention, particularly the edge-of-stability phenomenon [8]. Lu et al. [37] investigates the phenomenon "benign oscillation", which suggests that SGD with a large learning rate can effectively help neural networks learn weak features thereby benefiting generalization. Several works [20, 48, 11] show that label noise with large step size has a sparcifying effect for sparse linear regression. This paper instead studies multi-environment scenarios and fills in the understanding of the impact of randomness on matrix sensing problems.

**Federated Learning.** Federated learning [39, 26] is a machine learning paradigm where data is stored separately and locally on multiple clients and not exchanged, and clients collaboratively train a model. Extensive work has focused on designing effective decentralized algorithms (e.g. [39, 29]) while preserving privacy (e.g. [10, 7]). The importance of fairness in federated learning has also garnered attention [30, 33]. One important issue in federated learning is to handle the heterogeneity across the data and hardware. Our work shows that by training with certain stochastic gradient descent methods, the system can automatically remove the bias from the individual environment and thus learn the invariant features. Our work provides insights into discovering the implicit regularization effects of standard decentralized algorithms.

**Invariance Learning.** This research line initiates from causal inference literature [43, 40, 15] since invariant covariates correspond to *direct cause*. From theoretic aspects, Fan et al. [13] proposes the EILLS method that provably achieves invariant variable selections under mild conditions for linear models. Invariance learning has raised much attention in machine learning since Arjovsky et al. [2] proposes the structural-agnostic framework IRM. Subsequent works analyze its limitations [44, 28] or propose variant methods [50, 36, 34, 35, 21, 51] as regularization and reweighting. About the failure of classical methods, Wald et al. [49] construct a hard problem and show that interpolation-based methods fail to learn invariance.

To the best of our knowledge, all the existing works consider specific algorithms to realize invariance learning or constructing hard cases that classical methods fail. In contrast, this paper studies commonly used training algorithms and aims to understand how the algorithms can go beyond learning associations to achieve invariance learning in certain scenarios.

## 3 Main Results

### 3.1 Problem Formulation

**Data Generating Process.** Suppose we observe data from a set of environments $\mathcal{E}$ sequentially. Let $D$ be some distribution on $\mathcal{E}$. At each time $t = 0, 1, \ldots$, we receive $m$ samples $\{(\mathbf{X}_i^{(e_t)}, y_i^{(e_t)})\}_{i=1}^m \subset \mathbb{R}^{d \times d} \times \mathbb{R}$ from environment $e_t \sim D$ satisfying

$$y_i^{(e_t)} = \langle \mathbf{X}_i^{(e_t)}, \mathbf{A}^\star \rangle + \langle \mathbf{X}_i^{(e_t)}, \mathbf{A}^{(e_t)} \rangle, \quad i = 1, \ldots m, \tag{1}$$

where $\mathbf{A}^\star$ is an unknown rank $r_1$ $d \times d$ symmetric and positive definite matrix that represents the *true signal* invariant across different environments, $\mathbf{A}^{(e_t)}$ is an unknown $d \times d$ symmetric matrix with rank at most $r_2$ that represents the *spurious signal* that may vary. Here $\langle \mathbf{A}, \mathbf{B} \rangle = \text{trace}(\mathbf{B}^\top \mathbf{A})$. We aim to estimate the $\mathbf{A}^\star$ using data from heterogeneous environments.

**Algorithm.** We consider running batch gradient descent on an over-parametrization of the model, where at each step $t$ one gradient update is performed using the data from environment $e_t$. To be specific, we parameterize our fitted model as $y = \langle \mathbf{A}, \mathbf{U}\mathbf{U}^\top \rangle$ with a $d \times d$ matrix $\mathbf{U}$ for the sake of simplicity. One can generally use the parameterization $\mathbf{X} = \mathbf{U}\mathbf{U}^\top - \mathbf{V}\mathbf{V}^\top$ by the same technique of HaoChen et al. [20], Fan et al. [14]. We initialize $\mathbf{U}$ as $\mathbf{U}_0 = \alpha \mathbf{I}_d$ for some small enough $\alpha > 0$. At timestep $t$, we run a one-step gradient descent on the standard least squares loss using $\{(\mathbf{X}_i^{(e_t)}, y_i^{(e_t)})\}_{i=1}^m$:

$$L_t(\mathbf{U}) = \frac{1}{2m} \sum_{i=1}^m \left( y_i^{(e_t)} - \langle \mathbf{X}_i^{(e_t)}, \mathbf{U}\mathbf{U}^\top \rangle \right)^2. \tag{2}$$

That is, $\mathbf{U}_0 = \alpha \mathbf{I}_d$ and

$$\mathbf{U}_{t+1} = \mathbf{U}_t - \eta \nabla L_t(\mathbf{U}_t) = \left( \mathbf{I}_d - \eta \frac{1}{m} \sum_{i=1}^m (\langle \mathbf{X}_i^{(e_t)}, \mathbf{U}_t \mathbf{U}_t^\top \rangle - y_i^{(e_t)}) \mathbf{X}_i^{(e_t)} \right) \mathbf{U}_t \tag{3}$$

---

**Algorithm 3** `HeteroSGD`

---

Set $\mathbf{U}_0 = \alpha \mathbf{I}_d$, where $\alpha$ is a small positive constant to be determined later.
Set large step size $\eta = \Theta(1)$.
**for** $t = 1, \ldots, T - 1$ **do**
    Receive $m$ samples $\{(\mathbf{X}_i^{(e_t)}, y_i^{(e_t)})\}_{i=1}^m$ from current environment $e_t$.
    Gradient Descent $\mathbf{U}_{t+1} = \mathbf{U}_t - \frac{\eta}{m} \left[ \sum_{i=1}^m \left( \langle \mathbf{X}_i^{(e_t)}, \mathbf{U}_t \mathbf{U}_t^\top \rangle - y_i^{(e_t)} \right) \mathbf{X}_i^{(e_t)} \right] \mathbf{U}_t$.
**end for**
**Output:** $\mathbf{U}_T$.

---

for $t = 0, \ldots, T - 1$. See a complete presentation in Algorithm 3.

The algorithm adopts a constant level step size $\eta$ and $\log(\alpha^{-1})$ level number of iterations $T$, i.e. $\eta = \Theta(1)$ and $T = \Theta(\log(\alpha^{-1}))$, and use $\mathbf{U}_T \mathbf{U}_T^\top$ as our estimate of $\mathbf{A}^\star$.

**Standard Method: Pooled Stochastic Gradient Descent.** As a comparison, we consider the standard approach where data in each batch come from different environments and the weights follow from $D$. To be specific, the pooled stochastic gradient descent over all environments adopted the update rule

$$\mathbf{U} \leftarrow \mathbf{U} - \eta \nabla \bar{\mathcal{L}}(\mathbf{U}), \text{ where } \bar{\mathcal{L}}(\mathbf{U}) = \frac{1}{2m} \sum_{i=1}^m \left[ \left( y_i^{(e_i)} - \langle \mathbf{X}_i^{(e_i)}, \mathbf{U}\mathbf{U}^\top \rangle \right)^2 \right], \quad e_i \sim D. \quad (4)$$

## 3.2 Assumptions

We first impose some standard assumptions used in matrix sensing. Since we are dealing with learning true invariant signals from heterogeneous environments, several conditions on the structure of the invariant signal $\mathbf{A}^\star$ and the spurious signals $\mathbf{A}^{(e)}$ should be imposed.

**Assumption 1** (Invariant and Spurious Space). *There exists $\mathbf{U}^\star \in \mathbb{R}^{d \times r_1}$ and $\mathbf{V}^\star \in \mathbb{R}^{d \times r_2}$ both with orthogonal columns, i.e., $(\mathbf{U}^\star)^\top \mathbf{U}^\star = \mathbf{I}_{r_1}$ and $(\mathbf{V}^\star)^\top \mathbf{V}^\star = \mathbf{I}_{r_2}$ such that*

    *(a). $C \log^4(d) \leq r_1 \wedge r_2$ and $d \geq (r_1 + r_2)^C$ for some large absolute constant $C$.*

    *(b). $\mathbf{A}^\star = \mathbf{U}^\star (\mathbf{U}^\star)^\top$.*

    *(c). $\mathbf{A}^{(e)} = \mathbf{V}^\star \mathbf{\Sigma}^{(e)} (\mathbf{V}^\star)^\top$ with some symmetric $r_2 \times r_2$ matrix $\mathbf{\Sigma}^{(e)}$ for any $e \in \mathcal{E}$.*

    *(d). $\|(\mathbf{U}^\star)^\top \mathbf{V}^\star\| \leq \epsilon_1$ for some small quantity $\epsilon_1 \geq 0$.*

In Condition (b), we assume that the singular values of the true signal $\mathbf{A}^\star$ are the same to simplify the presentation since our main focus is to reduce the spurious signals. It holds for the basic case when there is only one invariant signal, i.e. $r_1 = 1$. The analysis for varying singular values using the technique of Li et al. [31] is deferred to Section D in Appendix. Other assumptions are usual and easy to achieve. Condition (a) requires that the total dimension of invariant signals and spurious signals are small relative to the ambient dimension $d$. Condition (c) resembles the RIP condition [6] in sparse feature selection [5]. Condition (d) says the overlap of invariant subspace and spurious subspace should be small. Such a condition can be easily satisfied for random projections in high dimensions where $r_1 + r_2 \ll d$, under which we have $\epsilon_1 = \Theta(\sqrt{(r_1 + r_2)/d})$, see Proposition 1 below.

**Proposition 1.** *Let $\mathbf{M}_1 \in \mathbb{R}^{d \times r_1}$ and $\mathbf{M}_2 \in \mathbb{R}^{d \times r_2}$ be two mutually independent random matrix with i.i.d. $N(0, 1)$ entries. Denote their QR decompositions as $\mathbf{M}_1 = \mathbf{U}_1^\star \mathbf{R}_1$ and $\mathbf{M}_2 = \mathbf{U}_2^\star \mathbf{R}_2$, respectively. Then there exists a universal constant $C_1 > 0$ such that*

$$\left\| (\mathbf{U}_1^\star)^\top \mathbf{U}_2^\star \right\| \leq t \sqrt{\frac{r_1 + r_2}{d}}, \quad (5)$$

*with probability at least $1 - 4\exp\left(-C_1^{-1}d\right) - 2\exp\left(-C_1^{-1}(r_1 + r_2)t^2\right)$.*

**Assumption 2** (Regularity on Spurious Signal $\mathbf{\Sigma}^{(e)}$). *There exists some constant-level quantity $M_1, M_2$ such that*

$$\sup_{e \in \mathcal{E}, i \in [r_2]} |\mathbf{\Sigma}_{ii}^{(e)}| < M_1 \qquad \text{and} \qquad \min_{i \in [r_2]} \frac{\mathrm{Var}_{e \sim D}[\mathbf{\Sigma}_{ii}^{(e)}]}{1 + \left| \mathbb{E}_{e \sim D}[\mathbf{\Sigma}_{ii}^{(e)}] \right|} > M_2, \tag{6}$$

*where $M_1 < C_0 M_2$ for some universal constant $C_0 > 0$. Moreover, $\mathbf{\Sigma}^{(e)}$ is strongly diagonal dominant for any $e \in \mathcal{E}$, i.e.,*

$$\sup_{e \in \mathcal{E}} \max_{i \in [r_2]} r_2^2 \sum_{j \neq i} |\mathbf{\Sigma}_{ij}^{(e)}| \leq \frac{c_o}{M_2^{1.5}} \tag{7}$$

*where $c_o > 0$ is some universal constant.*

The first inequality in (6) requires that all the spurious signals have a uniform bound, under which a fixed step size can be adopted. The second inequality in (6) requires that the heterogeneity of the spurious signals be large compared to the bias of the spurious signals. For example, some variables receive different interventions in different environments. The condition (7) is imposed to prevent the explosion of spurious signals during training. When the diagonal and off-diagonal elements are of the same order, empirical studies and theoretical analyses in some toy examples illustrate the failure of recovering $\mathbf{A}^\star$. Condition (d) in Assumption 1 and (6) resemble the RIP condition in sparse feature selection. Example 1 can fulfill all our conditions.

Finally, we impose assumptions on measurements. Recall the RIP condition [6]:

**Definition 1** (RIP for Matrices [6]). *A set of linear measurements $\mathbf{X}_1, \ldots, \mathbf{X}_m$ satisfy the restricted isometry property (RIP) with parameter $(s, \delta)$ if the following inequality*

$$(1 - \delta) \|\mathbf{M}\|_F^2 \leq \frac{1}{m} \sum_{i=1}^m \langle \mathbf{X}_i, \mathbf{M} \rangle^2 \leq (1 + \delta) \|\mathbf{M}\|_F^2 \tag{8}$$

*holds for any $d \times d$ matrix $\mathbf{M}$ with rank at most $s$.*

**Assumption 3** (RIP Condition for Linear Measurements). *$\mathbf{X}_1^{(e_t)}, \ldots, \mathbf{X}_m^{(e_t)}$ satisfies the RIP with parameter $s = 4(r_1 + r_2)$ and $\delta \lesssim \frac{1}{(M_2 \log(d))^{1.5} r_2^{2.5} \sqrt{r_1 + r_2}}$ for all $e \in \mathcal{E}$.*

It is known from Candès and Plan [6] that for symmetric Gaussian measurements, sample complexity $m = \tilde{\Omega}(ds\delta^{-2}, M_2) = d \, \mathrm{poly}(r, \log(d)) \ll d^2$ suffices.

### 3.3 Convergence Analysis

The main conceptual challenge in the problem is that any $\mathbf{U}$ with $\mathbf{U}\mathbf{U}^\top = \mathbf{A}^\star$ is no longer a local minimum since $\mathbb{E}_{e \sim D}[\mathbf{\Sigma}^{(e)}]$ is non-zero and could even be comparable to $\mathbf{A}^\star$. This further implies that running stochastic gradient descent on pooled data will fail to recover $\mathbf{A}^\star$. However, our main result below shows that simply adopting online gradient descent with "heterogeneous batches" can successfully recover the true, invariant signal from heterogeneous environments.

**Theorem 2** (Main Theorem). *Under Assumption 1-3, suppose further that $\epsilon_1 < \delta/2$. Define $\delta^\star := (c_v M_2 \log(d))^{1.5} \delta r_2^2 \sqrt{r_1 + r_2}$ for some absolute constant $c_v$. If we choose $\eta \in (24 M_2^{-1}, \frac{1}{64} M_1^{-1})$ and $\alpha \in (1/d^4, 1/d^2)$, then running Algorithm 3 in $T = \Theta(\log(\alpha^{-1})/\eta)$ steps, the algorithm outputs $\mathbf{U}_T$ that satisfies*

$$\|\mathbf{U}_T \mathbf{U}_T^\top - \mathbf{A}^\star\|_F \leq C \max\{\delta^{\star 2} \sqrt{r_1} M_1^2, \delta^\star M_1\} \log^2 d \tag{9}$$

*for some absolute constant $C$, with probability over $0.99$.*

Consider the case where $r_1, r_2, M_1$ are sufficiently large but is regarded at constant level, and the batch size $m$, ambient dimension $d$ satisfy $d \log^2(d) \ll m$. It follows from the RIP result [6] with $\delta = \Theta(\sqrt{d/m})$ and Theorem 2 that one can adopt $\alpha = \Theta(d^{-1})$, $\eta = \Theta(1)$, and early stop $T = \Theta(\log d)$ such that

$$\mathbb{P}\left[ \|\mathbf{U}_T \mathbf{U}_T^\top - \mathbf{A}^\star\|_F \leq C_1 \log(d)\sqrt{d/m} \right] \geq 1 - C_1 (d \log(d)/m)^{2/5} \tag{10}$$

provided $\epsilon_1 \le C_1^{-1}\sqrt{d/m}$ with some large enough constant $C_1 > 0$. In this case, it follows from (10) that one can distinguish the true invariant signals from those spurious heterogeneous ones since

$$\max\left\{\left\|(\mathbf{U}^\star)^\top \mathbf{U}_T \mathbf{U}_T^\top \mathbf{U}^\star - \mathbf{I}_{r_1}\right\|_2, \left\|(\mathbf{V}^\star)^\top \mathbf{U}_T \mathbf{U}_T^\top \mathbf{V}^\star\right\|_2\right\} = o_{\mathbb{P}}(1). \tag{11}$$

The underlying reason why the online gradient descent can recover $\mathbf{A}^\star$ is that the heterogeneity of $\mathbf{A}^{(e)}$ and the randomness in the SGD algorithm jointly prevent it from moving in the direction of spurious signals. At the same time, the standard RIP conditions and the almost orthogonality between $\mathbf{U}^\star$ and $\mathbf{V}^\star$ in Condition 1 ensure a steady movement towards the invariant signals.

Conversely, running pooled stochastic gradient descent using all data will result in a biased solution:

**Theorem 3** (Negative Result for Pooled SGD). *Under the assumptions of Theorem 2 and some mild conditions, for the certain case where $\mathbf{U}^\star \perp \mathbf{V}^\star$ and $\mathbb{E}_{e \in D}\mathbf{\Sigma}^{(e)} = \mathbf{I}_{r_2}$, if we perform SGD over all samples with batch size $m = \Omega(d\operatorname{poly}(r_1 + r_2, M_1 M_2, \log(d)))$ and ends with $T = \Theta(\log d)$, then $\mathbf{U}_t$ keeps approaching $\mathbf{U}^\star \mathbf{U}^{\star\top} + \mathbf{V}^\star \mathbf{V}^{\star\top}$, in the sense that*

$$\left\|\mathbf{U}_T \mathbf{U}_T^\top - \mathbf{U}^\star \mathbf{U}^{\star\top} - \mathbf{V}^\star \mathbf{V}^{\star\top}\right\|_F \le o(1), \tag{12}$$

*during which for all $t = 0, 1, \ldots, T$:*

$$\left\|\mathbf{U}_t \mathbf{U}_t^\top - \mathbf{A}^\star\right\|_F \gtrsim \sqrt{r_1 \wedge r_2}. \tag{13}$$

The convergence (12) is similar to (9) in derivation. To see this, since each update uses batch from the whole data, the update in effect degenerates to the case for one environment with no heterogeneity. Now the one-environment invariant solution $\mathbf{A}^\star$ in (9) is exactly equal to $\mathbf{U}^\star \mathbf{U}^{\star\top} + \mathbf{V}^\star \mathbf{V}^{\star\top}$ in (12). One can also show that for sufficiently large $t$, $\mathbf{U}_t \mathbf{U}_t^\top$ is sufficiently away from $\mathbf{A}^\star$, indicating that the biased estimation is not attributed to early stopping.

Our framework can be applied to learning the invariant features for a two-layer neural network with quadratic activation functions, by recognizing the fact that [31]:

$$\mathbf{1}^\top q(\mathbf{U}\mathbf{x}) = \left\langle \mathbf{x}\mathbf{x}^\top, \mathbf{U}\mathbf{U}^\top \right\rangle, \tag{14}$$

where $q(\cdot)$ is the element-wise quadratic function. The following example shows that Theorem 7 implies success of invariant feature learning for 2-layer NN when the ground truth invariant and variant features are independent random vectors sampled from normal distribution.

**Example 1** (Two-Layer NN with Quadratic Activation). *Let $\mathbf{a}_1, \cdots, \mathbf{a}_r \in \mathbb{R}^d$ be random vectors sampled from normal distribution $N(0, \frac{1}{d}\mathbf{I}_d)$. For environment $e \in \mathcal{E}$, suppose the target function is determined by $r_1$ invariant features and $r_2$ variant admits that for each sample $(\mathbf{x}_i^{(e)}, y_i^{(e)})$:*

$$y_i^{(e)} = \sum_{j=1}^{r_1} q(\mathbf{a}_j^\top \mathbf{x}_i^{(e)}) + \sum_{j=r_1+1}^{r} a_j^{(e)} q(\mathbf{a}_j^\top \mathbf{x}_i^{(e)}) = \left\langle \mathbf{x}_i^{(e)} \mathbf{x}_i^{(e)\top}, \sum_{j=1}^{r_1} \mathbf{a}_j \mathbf{a}_j^\top + \sum_{j=r_1+1}^{r} a_j^{(e)} \mathbf{a}_j \mathbf{a}_j^\top \right\rangle, \tag{15}$$

*which is equivalent to matrix sensing problem with*

$$\mathbf{A}^\star = \sum_{j=1}^{r_1} \mathbf{a}_j \mathbf{a}_j^\top, \quad \mathbf{A}^{(e)} = \sum_{j=r_1+1}^{r} a_j^{(e)} \mathbf{a}_j \mathbf{a}_j^\top \text{ and } \mathbf{X}_i^{(e)} = \mathbf{x}_i^{(e)} \mathbf{x}_i^{(e)\top}. \tag{16}$$

*And our goal is to train a two-layer NN to capture the invariant features $(\mathbf{a}_1, \ldots, \mathbf{a}_{r_1})$. In this example, the invariant component and the spurious component have a more intuitive characterization: they are two disjoint groups of neurons. Moreover, it can be shown that the invariant and variant features are nearly orthogonal (Proposition 1). Then if $\{a_j^{(e)}\}_{j,e}$ satisfies $\frac{\sup_{e,j}\{|a_j^{(e)}|\} \cdot \max_j\{1+|\mathbb{E}_e a_j^{(e)}|\}}{\min_j\{\operatorname{Var}_e[a_j^{(e)}]\}} < c_0$ for some absolute constant $c_0$, the variant version of Algorithm 3 returns a solution that only significantly selects invariant features with probability over $0.99$. See Section C and Theorem 7 for details.*

## 4 Proof Sketch

We define the invariant part $\mathbf{R}_t \in \mathbb{R}^{d \times r_1}$, spurious part $\mathbf{Q}_t \in \mathbb{R}^{d \times r_2}$ in $\mathbf{U}_t$ as

$$\mathbf{R}_t := \mathbf{U}_t^\top \mathbf{U}^\star \quad \text{and} \quad \mathbf{Q}_t := \mathbf{U}_t^\top \mathbf{V}^\star \tag{17}$$

and let the residual be the error part, that is,

$$\mathbf{E}_t := \mathbf{U}_t - \left(\mathbf{U}^\star \mathbf{R}_t^\top + \mathbf{V}^\star \mathbf{Q}_t^\top\right) = (\mathbf{I} - \mathbf{U}^\star \mathbf{U}^{\star\top} - \mathbf{V}^\star \mathbf{V}^{\star\top})\mathbf{U}_t. \tag{18}$$

It is worth noticing that $\mathrm{Id}_{\mathbf{U}^\star} = \mathbf{U}^\star \mathbf{U}^{\star\top}$ and $\mathrm{Id}_{\mathbf{V}^\star} = \mathbf{V}^\star \mathbf{V}^{\star\top}$ are both orthogonal projections, and $\mathrm{Id}_{\mathrm{res}} := \mathbf{I} - \mathrm{Id}_{\mathbf{U}^\star} - \mathrm{Id}_{\mathbf{V}^\star}$ is not.

It follows from the model (1) and the gradient update that

$$\mathbf{U}_{t+1} = \mathbf{U}_t - \eta \frac{1}{m} \sum_{i=1}^m \langle \mathbf{X}_i^{(e_t)}, \mathbf{U}_t \mathbf{U}_t^\top - \mathbf{A}^\star - \mathbf{A}^{(e_t)} \rangle \mathbf{X}_i^{(e_t)} \mathbf{U}_t. \tag{19}$$

We use operator $\mathsf{E}_{e_t} \circ (\mathbf{M}) \in \mathbb{R}^{d \times d}$ to denote the RIP error of the batch at time step $t$ for some $d \times d$ matrix $\mathbf{M}$, i.e.,

$$\mathsf{E}_{e_t} \circ (\mathbf{M}) := \frac{1}{m} \sum_{i=1}^m \langle \mathbf{X}_i^{(e_t)}, \mathbf{M} \rangle \mathbf{X}_i^{(e_t)} - \mathbf{M}. \tag{20}$$

We also write $\mathsf{E}_t \circ (\mathbf{M}) := \mathsf{E}_{e_t} \circ (\mathbf{M})$ when there is no ambiguity, and we simply denote matrix $\mathsf{E}_t = \mathsf{E}_t \circ \left(\mathbf{U}_t \mathbf{U}_t^\top - \mathbf{U}^\star \mathbf{U}^{\star\top} - \mathbf{V}^\star \mathbf{\Sigma}_t \mathbf{V}^{\star\top}\right)$ with $\mathbf{\Sigma}_t := \mathbf{\Sigma}^{(e_t)}$. Then the gradient update of $\mathbf{U}_t$ can be written as

$$\mathbf{U}_{t+1} = \mathbf{U}_t - \eta \left(\mathbf{U}_t \mathbf{U}_t^\top - \mathbf{U}^\star \mathbf{U}^{\star\top} - \mathbf{V}^\star \mathbf{\Sigma}_t \mathbf{V}^{\star\top}\right) \mathbf{U}_t - \eta \underbrace{\mathsf{E}_t \mathbf{U}_t}_{\text{RIP Error}}. \tag{21}$$

Combining our definition (17) with (21), we obtain

$$\begin{aligned}
\mathbf{R}_{t+1} &= \left(\mathbf{U}_t - \eta(\mathbf{U}_t \mathbf{U}_t^\top - \mathbf{U}^\star \mathbf{U}^{\star\top} - \mathbf{V}^\star \mathbf{\Sigma}_t \mathbf{V}^{\star\top})\mathbf{U}_t - \eta \mathsf{E}_t \mathbf{U}_t\right)^\top \mathbf{U}^\star \\
&= \underbrace{(\mathbf{I} - \eta \mathbf{U}_t^\top \mathbf{U}_t + \eta \mathbf{I})\mathbf{R}_t}_{\text{Dominating Dynamics}} + \eta \underbrace{\mathbf{U}_t^\top \mathbf{V}^\star \mathbf{\Sigma}_t \mathbf{V}^{\star\top} \mathbf{U}^\star}_{\text{Interaction Error}} - \eta \underbrace{\mathbf{U}_t^\top \mathsf{E}_t \mathbf{U}^\star}_{\text{RIP Error}}.
\end{aligned} \tag{22}$$

$$\begin{aligned}
\mathbf{Q}_{t+1} &= \left(\mathbf{U}_t - \eta(\mathbf{U}_t \mathbf{U}_t^\top - \mathbf{U}^\star \mathbf{U}^{\star\top} - \mathbf{V}^\star \mathbf{\Sigma}_t \mathbf{V}^{\star\top})\mathbf{U}_t - \eta \mathsf{E}_t \mathbf{U}_t\right)^\top \mathbf{V}^\star \\
&= \underbrace{\mathbf{Q}_t - \eta \mathbf{U}_t^\top \mathbf{U}_t \mathbf{Q}_t + \eta \mathbf{Q}_t \mathbf{\Sigma}_t}_{\text{Fluctuation Dynamics}} + \eta \underbrace{\mathbf{U}_t^\top \mathbf{U}^\star \mathbf{U}^{\star\top} \mathbf{V}^\star}_{\text{Interaction Error}} - \eta \underbrace{\mathbf{U}_t^\top \mathsf{E}_t \mathbf{V}^\star}_{\text{RIP Error}}.
\end{aligned} \tag{23}$$

For the error part, combining (18) with (21) yields

$$\begin{aligned}
\mathbf{E}_{t+1} &= \mathrm{Id}_{\mathrm{res}} \left(\mathbf{U}_t - \eta(\mathbf{U}_t \mathbf{U}_t^\top - \mathbf{U}^\star \mathbf{U}^{\star\top} - \mathbf{V}^\star \mathbf{\Sigma}_t \mathbf{V}^{\star\top})\mathbf{U}_t - \eta \mathsf{E}_t \mathbf{U}_t\right) \\
&= \underbrace{\mathbf{E}_t(\mathbf{I} - \eta \mathbf{U}_t^\top \mathbf{U}_t)}_{\text{Shrinkage Dynamics}} + \eta \underbrace{\mathrm{Id}_{\mathrm{res}}(\mathbf{U}^\star \mathbf{U}^{\star\top} + \mathbf{V}^\star \mathbf{\Sigma}_t \mathbf{V}^{\star\top})\mathbf{U}_t}_{\text{Interaction Error}} - \eta \underbrace{\mathrm{Id}_{\mathrm{res}} \mathsf{E}_t \mathbf{U}_t}_{\text{RIP Error}}.
\end{aligned} \tag{24}$$

For the invariant part $\mathbf{R}_t$, though different singular values of $\mathbf{R}_t$ will grow at different speeds because of the randomness from RIP error and $e_t$, we claim that all the singular values of $\mathbf{R}_t$ are close to $\mathrm{R}_t$ during the training process, where the scalar sequence $\mathrm{R}_t$ is defined recursively as

$$\mathrm{R}_{t+1} = (1 - \eta \mathrm{R}_t^2 + \eta)\mathrm{R}_t, \quad \mathrm{R}_0 = \alpha. \tag{25}$$

The dynamics of $\mathbf{Q}_t$ are very complicated because of the randomness of $e_t$ and the RIP error. Such a dynamic will also impact that of $\mathbf{R}_t$ and $\mathbf{E}_t$ through the complicated dependencies between these three parts, which will also make it difficult to utilize probability inequalities applicable under independence. Instead, we claim that such a "fluctuation dynamics" of $\mathbf{Q}_t$ can be controlled as

$$\|\mathbf{Q}_t\| < \mathrm{poly}(\log(d), r, M_1)\mathrm{L}_t \ \text{ with } \ \mathrm{L}_t = \begin{cases} \alpha & , t < O(\frac{1}{\eta}\log(r_1 + r_2)) \\ O(\delta M_1 \sqrt{r_1 + r_2}\mathrm{R}_t) & , t \geq O(\frac{1}{\eta}\log(r_1 + r_2)) \end{cases}. \tag{26}$$

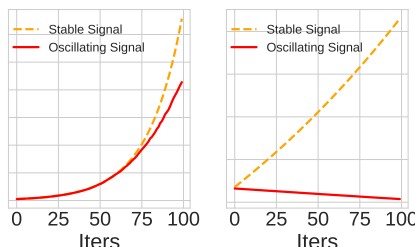

We now offer an informal illustration for how oscillations "shrink" spurious signal. We simply omit error terms. When matrix $\mathbf{\Sigma}^{(t+1)}$ is diagonal, from (23), the $i$-th column $\mathbf{q}_t$ of $\mathbf{Q}_t$ should satisfies: $\|\mathbf{q}_{t+1}\| \leq (1 + \eta\mathbf{\Sigma}_{ii}^{(t+1)})\|\mathbf{q}_t\|$. Let $\xi \stackrel{\text{def}}{=} \mathbf{\Sigma}_{ii}^{(t+1)}$ and assume $M \geq |\xi|$ a.s.. Introduce a **concave** function [20] $\phi(x) = x^\gamma, \gamma \in (0,1)$. When $\eta M < \frac{1}{16}$, do second-order Taylor's expansion at $\phi(1)$ in $(a)$ below:

Figure 2: The left figure shows $\mathbb{E}\|\mathbf{q}_t\|$ and the right figure shows $\mathbb{E}\|\mathbf{q}_t\|^{0.1}$.

$$\mathbb{E}\Big[\phi(\|\mathbf{q}_{t+1}\|)\Big] \leq \mathbb{E}\Big[\phi(1 + \eta\xi)\Big] \cdot \phi(\|\mathbf{q}_t\|)$$
$$\stackrel{(a)}{\approx} \Big(1 + \eta\gamma\mathbb{E}[\xi] - \frac{\eta^2\gamma(1-\gamma)}{2}\operatorname{Var}[\xi]\Big)\phi(\|\mathbf{q}_t\|) < \phi(\|\mathbf{q}_t\|),$$

where $\mathbb{E}[\cdot]$ is w.r.t. $\xi$. So spurious signal keeps small when $\frac{1}{16M} > \eta > \frac{4\mathbb{E}[\xi]}{(1-\gamma)\operatorname{Var}[\xi]}$. See the figure for illustration in Figure 2. While $\mathbb{E}[\|\mathbf{q}_t\|]$ (shown in the left figure) increases since the signals have positive expectations, $\mathbb{E}[\|\mathbf{q}_t\|^{0.1}]$ (shown in the right figure) decreases. Note that the above intuition is informal and the formal argument is deferred in Lemma 6 and Lemma 7 in Appendix.

The entire training process can be divided into two phases. In Phase 1, the invariant signals $\mathbf{R}_t$ increase rapidly while the spurious signals $\mathbf{Q}_t$ fluctuate but remain at a low level. Phase 1 ends in $O(\frac{1}{\eta}\log(\frac{1}{\alpha}))$ steps when $\mathbf{R}_t$ attains $\Theta(1)$-order (see Theorem 4). In Phase 2, the magnitudes of $\mathbf{Q}_t$ and $\mathbf{E}_t$ stay low, while all the singular values of $\mathbf{R}_t$ approach 1 (See Theorem 5). We defer the details to Appendix.

## 5   Simulations

In this section, we present our simulations. We design three sets of experiments. In the first set of experiments, we show with the growth of environment heterogeneity, invariance learning is achievable. For the second set of experiments, we show that given heterogeneous data, invariance learning is achievable with the growth of step size[3]. For the third set of experiments, we compare `HeteroSGD` (Algorithm 3) and Pooled SGD. In Section B.2 we also perform simulations for Pooled SGD with small batch size.

In below two sets of experiments, we set the scale of initialization $\alpha = 10^{-3}$, problem dimension $d = 100$, $r_1 = 1$ and $r_2 = 1$. Let the true signal be $\mathbf{A}^\star = \mathbf{u}\mathbf{u}^\top$. Denote the heterogeneity parameter by $M$. The environment is generated by $\mathbf{A}^{(e)} = \mathbf{A}^\star + s^{(e)}\mathbf{v}\mathbf{v}^\top$ where $s^{(e)} \sim \operatorname{Unif}\{1-M, 1+M\}$, and the default of $\eta$ is 0.05. The number of linear measurements is set to be $m = 8000$ with elements following from i.i.d $N(0,1)$. For the third sets, we set $(r_1, r_2, d, \mathbb{E}s^{(e)}) = (3, 2, 40, 0.5)$, $m = 2800$ for `HeteroSGD` and $m = 5600$ without replacement for Pooled SGD. The plots show signal recovery proportion, 1.0 indicates fully recovery.

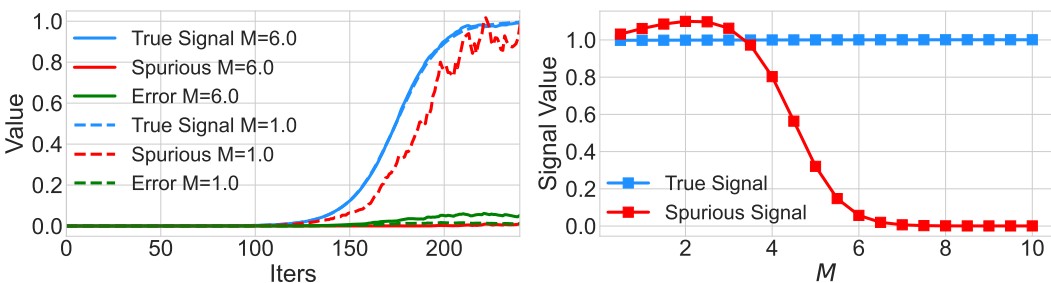

Figure 3: The left figure shows that the heterogeneity facilitates us to eliminate the spurious signal and learn the invariance. The right figure shows that both true and spurious signals flow up when $M$ is small, the "phase transition" happens around $M = 5$.

---

[3] A smaller step size can reduce the noise arising from heterogeneity, making the dynamics more similar to those of Gradient Descent.

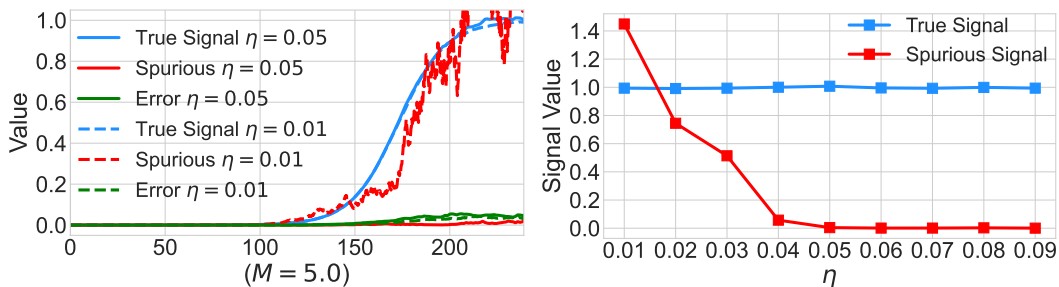

Figure 4: The left figure shows that the large step size helps eliminate the spurious signal. The right figure shows that both true and spurious signals flow up when $\eta$ is small, and when $\eta \geq 0.05$, the spurious signal is eliminated.

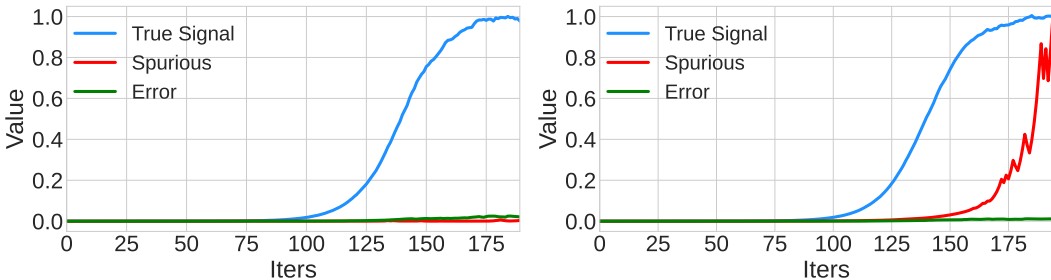

Figure 5: The left figure shows that heterogeneity helps eliminate the spurious signal. The right figure shows that Pooled SGD fits invariant signal and spurious signal simultaneously without distinction.

## 6 Conclusions

This paper explains that implicit bias of heterogeneity leads the model learning towards invariance and causality. We show that under heterogeneous environments, online gradient descent with large step sizes can select out the invariant matrix in the over-parameterized matrix sensing models. We conjecture that both heterogeneity and stochasticity are indispensable. Over-parameterization may not be. We leave future studies to understand the necessity of the three factors.

## 7 Acknowledgement

C. Fang was supported by National Key R&D Program of China (2022ZD0114902), the NSF China (No.62376008).

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

# Contents

## A   Deferred Proofs in Theorem 2

This section is organized as follows: In Section A.1, we state some useful properties from the definition of RIP. In Section A.2 and A.3, we formally define the auxiliary sequences we use to control the dynamics and develop several useful lemmas we frequently use. In Section A.4 and A.5, we bound $\mathbf{Q}_t$ and $\mathbf{E}_t$ respectively. In Section A.6 and A.7, we prove Theorem 4 and Theorem 5.

### A.1   Restricted Isometry Properties

In this section, we list some useful implications of the definition of RIP property. Below we assume the set of linear measurements $\mathbf{A}_1^{(e_t)}, \ldots, \mathbf{A}_m^{(e_t)} \in \mathbb{R}^{d \times d}$ satisfy the RIP property with parameter

$(r, \delta)$ and denote $\mathsf{E}_t \circ (\mathbf{M}) := \frac{1}{m} \sum_{i=1}^{m} \langle \mathbf{X}_i^{(e_t)}, \mathbf{M} \rangle \mathbf{X}_i^{(e_t)} - \mathbf{M}$ for some symmetric $d \times d$ matrix $\mathbf{M}$. Some lemmas are direct corollaries and some lemmas serve as extensions to rank above $r$ case. The proof of these lemmas can be found in Li et al. [31].

**Lemma 1.** *Under the assumption of this subsection, if $\mathbf{X}, \mathbf{Y}$ are $d \times d$ matrices with rank at most $r$, then*

$$|\langle \mathsf{E}_t \circ (\mathbf{X}), \mathbf{Y} \rangle| \leq \delta \|\mathbf{X}\|_F \|\mathbf{Y}\|_F. \tag{27}$$

**Lemma 2.** *Under the assumption of this subsection, if $\mathbf{X}$ id $d \times d$ matrix with rank at most $r$ and $\mathbf{Z}$ is a $d \times d'$ matrix, then*

$$\|\mathsf{E}_t \circ (\mathbf{X})\mathbf{Z}\| \leq \delta \|\mathbf{X}\|_F \|\mathbf{Z}\|. \tag{28}$$

**Lemma 3.** *Under the assumption of this subsection, if $\mathbf{X}, \mathbf{Y}$ are $d \times d$ matrices and $\mathbf{Y}$ has rank at most $r$, then*

$$|\langle \mathsf{E}_t \circ (\mathbf{X}), \mathbf{Y} \rangle| \leq \delta \|\mathbf{X}\|_* \|\mathbf{Y}\|_F. \tag{29}$$

**Lemma 4.** *Under the assumption of this subsection, if $\mathbf{X}$ id $d \times d$ matrix and $\mathbf{Z}$ is a $d \times d'$ matrix, then*

$$\|\mathsf{E}_t \circ (\mathbf{X})\mathbf{Z}\| \leq \delta \|\mathbf{X}\|_* \|\mathbf{Z}\|. \tag{30}$$

Lemma 1 is from Candes [4]. The other three lemma can be derived from Lemma 1 through selecting $\mathbf{Z}$ or decomposing $\mathbf{X}$ into a series of rank-1 matrices [31].

## A.2 Additional Auxiliary Sequences

In this section, we additionally define some auxiliary sequences. Some for calibrating the dynamics, that is, describe how the dynamic progresses without error or randomness and track the trajectories with the accumulation of error. Some are used for characterizing the impact of randomness on the dynamic.

The next two deterministic sequences help to track the dynamic of singular values of $\mathbf{R}_t$ when it accumulates errors in each step.

**Definition 2.** *We define the following two deterministic sequences:*

$$\overline{\mathbf{R}}_{t+1} = (1 - \eta \overline{\mathbf{R}}_t^2 + \eta)\overline{\mathbf{R}}_t + \frac{\eta}{32} \log^{-1}\left(\frac{1}{\alpha}\right) \overline{\mathbf{R}}_t, \qquad \overline{\mathbf{R}}_0 = \alpha$$

$$\underline{\mathbf{R}}_{t+1} = (1 - \eta \underline{\mathbf{R}}_t^2 + \eta)\underline{\mathbf{R}}_t - \frac{\eta}{32} \log^{-1}\left(\frac{1}{\alpha}\right) \overline{\mathbf{R}}_t, \qquad \underline{\mathbf{R}}_0 = \alpha. \tag{31}$$

The next lemma shows that the deviation between $\underline{\mathbf{R}}_t$ and $\overline{\mathbf{R}}_t$ can be bounded.

**Lemma 5** (Bounded Deviation between $\underline{\mathbf{R}}_t$ and $\overline{\mathbf{R}}_t$). *Let the sequence $\mathbf{R}_t$ be defined as (25). Let $T_1$ be the first time $\mathbf{R}_t$ enters the region $(\frac{1}{3} - \eta, \frac{1}{3})$, we have*

$$\overline{\mathbf{R}}_t \leq (1 + 1/6)\mathbf{R}_t;$$
$$\underline{\mathbf{R}}_t \geq (1 - 1/6)\mathbf{R}_t, \tag{32}$$

*for any $t = 0, \dots, T_1$.*

*Proof.* Fist, for $\overline{\mathbf{R}}_t$ have that

$$\frac{\overline{\mathbf{R}}_{t+1}}{\mathbf{R}_{t+1}} = \frac{(1 - \eta \overline{\mathbf{R}}_t^2 + \eta)\overline{\mathbf{R}}_t + \frac{\eta}{32} \log^{-1}\left(\frac{1}{\alpha}\right) \overline{\mathbf{R}}_t}{(1 - \eta \mathbf{R}_t^2 + \eta)\mathbf{R}_t} \leq \left(1 + \frac{\eta}{32} \log^{-1}\left(\frac{1}{\alpha}\right)\right) \frac{\overline{\mathbf{R}}_t}{\mathbf{R}_t} \tag{33}$$

It takes $T_1 \leq \frac{4}{\eta} \log\left(\frac{1}{\alpha}\right)$ steps for $\mathbf{R}_t$ to reach $(\frac{1}{3} - \eta, \frac{1}{3})$. We can conclude that

$$\frac{\overline{\mathbf{R}}_{T_1}}{\mathbf{R}_{T_1}} \leq \left(1 + \frac{\eta}{32} \log^{-1}\left(\frac{1}{\alpha}\right)\right)^{T_1} \leq \exp(1/8) < 1 + \frac{1}{6} \tag{34}$$

where we use $1 - x \geq \exp(-2x), 1 + x \geq \exp(\frac{x}{2})$ for $x \in [0, 1/2]$. Similarly, For $\underline{\mathbf{R}}_t$, we have

$$\underline{\mathbf{R}}_{t+1} = (1 - \eta \underline{\mathbf{R}}_t^2 + \eta)\underline{\mathbf{R}}_t - \frac{\eta}{32} \log^{-1}\left(\frac{1}{\alpha}\right) \overline{\mathbf{R}}_t \geq (1 - \eta \underline{\mathbf{R}}_t^2 + \eta)\underline{\mathbf{R}}_t - \frac{\eta}{32} \cdot \frac{7}{6} \log^{-1}\left(\frac{1}{\alpha}\right) \mathbf{R}_t, \tag{35}$$

and

$$\frac{R_{t+1}}{\overline{R}_{t+1}} = \frac{(1 - \eta\underline{R}_t^2 + \eta)R_t - \frac{\eta}{32} \cdot \frac{7}{6}\log^{-1}\left(\frac{1}{\alpha}\right)R_t}{(1 - \eta R_t^2 + \eta)R_t} \geq \frac{\underline{R}_t}{R_t} - \frac{\eta}{32} \cdot \frac{7}{6}\log^{-1}\left(\frac{1}{\alpha}\right), \qquad (36)$$

which implies

$$\frac{\underline{R}_{T_1}}{\overline{R}_{T_1}} \geq 1 - \frac{7}{48} > 1 - \frac{1}{6}. \qquad (37)$$

One can also see that, $\overline{R}_t \leq (1 + 1/6)R_t$ and $\underline{R}_t \geq (1 - 1/6)R_t$ holds for any $t \leq T$, which completes our proof.

$\square$

Next, we formally define the calibration line $L_t$. In later parts, we can show that the norm of each column of $\mathbf{Q}_t$ behaves like a biased random walk with reflecting barrier $L_t$.

**Definition 3.** *Let $\alpha, R$ be defined as above. For $t = 0, 1, \ldots$, we define the calibration line:*

$$L_t = \alpha \vee 40M\delta\sqrt{r_1 + r_2}R_t. \qquad (38)$$

Next, we define a stochastic process $q_i^t$ based on $\mathbf{\Sigma}_t$. The reason why we define this sequence is that though the randomness only directly affects $\mathbf{Q}_t$, the dynamic of $\mathbf{E}_t$ and $\mathbf{R}_t$ also shares the randomness, therefore the dynamics become difficult to reason about since they are deeply coupled. Therefore, we define this "external" random sequence to dominate them.

**Definition 4** (Controller Sequence). *We fix the violation probability $p = c_v/(M_2\log(d))$ for some small absolute constant $c_v$. For each fixed $i$, we define a stochastic process $q_i^t$ for $t = 0, 1, 2, \ldots$ with $q_i^0 = \alpha$, and*

$$q_i^{t+1} = \begin{cases} q_i^t & , \quad \text{if there exists } \tau \leq t \text{ such that } q_i^\tau \geq p^{-1.5}r_2^{1.5} \cdot L_\tau \\ (1 + \eta\mathbf{\Sigma}_{ii}^{(t+1)} + 2\eta)q_i^t \vee L_{t+1} & , \quad \text{otherwise} \end{cases}$$

$q_i^t$ is used for providing an upper bound the norm of columns of $\mathbf{Q}_t$. Before $q_i^t$ hits the upper absorbing boundary $p^{-1.5}r_2^{1.5} \cdot L_t$, it can be considered as a "reflection and absorbing" process, with reflection barrier $L_t$ and absorbing barrier $p^{-1.5}r_2^{1.5} \cdot L_t$. The following lemma gives an upper bound for $\{q_i^t\}_{i,t}$:

**Lemma 6** (Upper bound for $q_i^t$). *With probability over $0.995$ over the randomness of the $\mathbf{\Sigma}^{(e_t)}$, for all $i = 1, 2, \ldots, r_2$ and $t = 0, 1, \ldots, T_2$, we have*

$$q_i^t < p^{-1.5}r_2^{1.5} \cdot L_t. \qquad (39)$$

To prove this, we define a family of random sequences $X_{k,t}^i$.

**Definition 5.** *For each $i = 1, \ldots, r_2$, we construct a family of non-negative stochastic processes $\{X_{k,t}^i\}_{t=0}^{T_2}$ for $k = 0, \ldots, T_2$ as follows:*

$$X_{k,t}^i = \begin{cases} L_t & , \quad 0 \leq t \leq k \leq T_2 \\ (1 + \eta\mathbf{\Sigma}_{ii}^{(e_t)} + 2\eta)X_{k,t-1}^i & , \quad 0 \leq k < t \leq T_2 \end{cases} \qquad (40)$$

$(X_{k,t}^i)_{k,t \in [T_2]}$ can be expressed as the following form:

$$\left(X_{k,t}^i\right) = \begin{pmatrix} L_0 & (1 + \eta\Sigma_{ii}^{(e_1)} + 2\eta)L_0 & \left(\prod_{s=1}^{2}(1 + \eta\Sigma_{ii}^{(e_s)} + 2\eta)\right) \cdot L_0 & \cdots & \left(\prod_{s=1}^{T}(1 + \eta\Sigma_{ii}^{(e_s)} + 2\eta)\right) \cdot L_0 \\ L_1 & L_1 & (1 + \eta\Sigma_{ii}^{(e_2)} + 2\eta)L_1 & \cdots & \left(\prod_{s=2}^{T}(1 + \eta\Sigma_{ii}^{(e_s)} + 2\eta)\right) \cdot L_1 \\ L_2 & L_2 & L_2 & \cdots & \left(\prod_{s=3}^{T}(1 + \eta\Sigma_{ii}^{(e_s)} + 2\eta)\right) \cdot L_2 \\ \vdots & \vdots & \vdots & \ddots & \\ L_{T_2} & L_{T_2} & L_{T_2} & \cdots & L_{T_2} \end{pmatrix}.$$

It can be noticed that $X_{k,t}^i$ and $q_i^t$ have close relations. At the beginning we have $q_i^t = X_{0,t}^i$, $t = 0, 1, \ldots$, progress along the 0-th row. If $q_i^t$ gets lower than the calibration line $L_t$ at some timestep $t = t_0$, it switches to the the $t_0$-th row $X_{t_0,t}^i, t = t_0, \ldots$ until the next time it gets lower than calibration line $L_t$, and so on. We can see that $q_i^t$ always progress along a certain row. Thus

$$\mathbb{P}\left(\exists k \text{ such that } q_i^t = X_{k,t}^i\right) = 1, \quad \forall i \in [r_2] \text{ and } t \in [T_2]. \tag{41}$$

Theerfore, any uniform bound of $X_{k,t}^i$ can also be a bound for $q_i^t$. Later in the context, we analyze $X_{k,t}^i$ for each $i$ so we omit the argument $i$ in $X_{k,t}^i$ for convenient notation.

We define $\sigma$-field $\mathcal{F}_t = \sigma(\mathbf{\Sigma}^{(e_0)}, \ldots, \mathbf{\Sigma}^{(e_{t-1})})$ for $t = 1, \ldots T_2$ and $\mathcal{F}_0 = \sigma(\emptyset)$. Then we have $\mathcal{F}_0 \subset \mathcal{F}_1 \subset \cdots \subset \mathcal{F}_{T_2}$ form a filtration. The next lemma shows that a certain power of $\{X_{k,t}\}_t$ is a non-negative supermartingale w.r.t. $\mathcal{F}_t$.

**Lemma 7.** *For each $i = 1, \ldots, r_2$ and $k = 0, \ldots, T_2$, if the learning rate $\eta$ satisfies $\eta \in (\frac{24}{M_2}, \frac{1}{64M_1})$, then the process $\{X_{k,t}^{2/3}\}_{t=0}^{T_2}$ is a non-negative supermartingale with respect to $\mathcal{F}_t$.*

*Proof.* First, its easy to verify the adaptiveness $X_{k,t}^{2/3} \in \mathcal{F}_t$ since $\mathbf{\Sigma}_{ii}^{(t)} \in \mathcal{F}_t$ for all $t = 0, 1, \ldots T_2$. Next, note that

$$\mathbb{E}\left[X_{k,t+1}^{2/3}|\mathcal{F}_t\right] = \begin{cases} X_{k,t}^{2/3} & , t + 1 \leq k \\ \mathbb{E}\left((1 + \eta(\mathbf{\Sigma}_{ii}^{(e_t)}) + 2\eta)\right)^{2/3} X_{k,t}^{2/3} & , t \geq k \end{cases} \tag{42}$$

So it suffices to prove that

$$\mathbb{E}_{e \sim D}\left[(1 + \eta\mathbf{\Sigma}_{ii}^{(e_t)} + 2\eta)^{2/3}\right] \leq 1.$$

For any $\gamma \in (0, 1)$ and $|x - 1| < \frac{1}{16}$, from Taylor's expansion, we have

$$x^{1-\gamma} \leq 1 + (1 - \gamma)(x - 1) - \frac{1}{4}(1 - \gamma)\gamma(x - 1)^2. \tag{43}$$

Therefore,

$$\mathbb{E}_{e_t}\left[(1 + \eta\mathbf{\Sigma}_{ii}^{(e_t)} + 2\eta)^{1-\gamma}\right] \leq 1 + \eta(1 - \gamma)(2 + \mathbb{E}\mathbf{\Sigma}_{ii}^{(e_t)}) - \frac{1}{4}\eta^2(1 - \gamma)\gamma \operatorname{Var}_{e_t}[\mathbf{\Sigma}_{ii}^{(e_t)}]. \tag{44}$$

Hence, it suffices to choose $\eta, \gamma$ such that

$$(2 + \mathbb{E}\mathbf{\Sigma}_{ii}^{(e_t)}) \leq \frac{1}{4}\eta\gamma \operatorname{Var}[\mathbf{\Sigma}_{ii}^{(e_t)}], \quad \eta < \frac{1}{64M_1}. \tag{45}$$

When $\gamma = \frac{1}{3}, \eta \in (\frac{24}{M_2}, \frac{1}{64M_1})$ suffices. Hence we prove $\mathbb{E}\left[(1 + \eta\mathbf{\Sigma}_{ii}^{(e_t)} + 2\eta)^{2/3}\right] \leq 1$ and we can conclude that

$$0 \leq \mathbb{E}\left[X_{k,t+1}^{2/3}|\mathcal{F}_t\right] \leq X_{k,t}^{2/3}. \tag{46}$$

$\square$

Now we are ready to prove Lemma 6.

*Proof of Lemma 6.* From the above observations, before $q_i^t$ hits the upper absorbing boundary $p^{-1.5}r_2^{1.5} \cdot L_t$, there always exists some $k$ such that $q_i^t = X_{k,t}$. Therefore, $q_i^t$ hits $p^{-1.5}r_2^{1.5} \cdot L_t$ implies there exists some $k$ that $X_{k,t}$ hits $p^{-1.5}r_2^{1.5} \cdot L_t$. So it suffices to bound $X_{k,t}$.

For any fixed $k = 0, \ldots, T_2$, we denote two stopping times:

$$\tau_k^0 \stackrel{\text{def}}{=} T_2 \wedge \min_{k \leq t \leq T_2}\{X_{k,t}^{2/3} < (L_t)^{2/3}\};$$
$$\tau_k^1 \stackrel{\text{def}}{=} T_2 \wedge \min_{k \leq t \leq T_2}\{X_{k,t}^{2/3} \geq (p^{-1.5}r_2^{1.5} \cdot L_t)^{2/3}\}. \tag{47}$$

One gets that

$$\mathbb{P}\left(\tau_k^1 < \tau_k^0\right) \overset{(a)}{\leq} \frac{1}{(p^{-1.5}r_2^{1.5} \cdot L_k)^{2/3}} \mathbb{E} X_{k,\tau_k^1 \wedge \tau_k^0}^{2/3} \overset{(b)}{\leq} \frac{1}{(p^{-1.5}r_2^{1.5} \cdot L_k)^{2/3}} \mathbb{E} X_{k,0}^{2/3}$$
$$\overset{(c)}{\leq} \frac{(L_k)^{2/3}}{(p^{-1.5}r_2^{1.5} \cdot L_t)^{2/3}} \leq (p^{-1.5}r_2^{1.5})^{-2/3}. \tag{48}$$

Where the inequality $(a)$ is from Markov's inequality. Inequality $(b)$ is from the optional stopping time theorem for supermartingales and inequality $(c)$ is from the fact that $L_t$ is non-decreasing. Therefore, we can conclude that:

$$\mathbb{P}(\exists i \leq r_2, \tau \leq T_2 \text{ such that } q_i^\tau \leq p^{-1.5}r_2^{1.5} \cdot L_\tau)$$
$$\leq r_2 \mathbb{P}(\exists k \text{ such that } \tau_k^1 < \tau_k^0)$$
$$\leq r_2 \sum_{k=0}^{T_2-1} \mathbb{P}\left(\tau_k^1 < \tau_k^0\right) \tag{49}$$
$$\leq r_2 T_2 (p^{-1.5}r_2^{1.5})^{-2/3}$$
$$\leq T_2 p$$

where the first inequality is simply a union bound over $i = 1, 2, ..., r_2$. Then

$$T_2 p \leq O(\eta^{-1}\log(d))\frac{c_v}{M_2 \log(d)} \leq O(M_2 \log(d))\frac{c_v}{M_2 \log(d)} \leq 0.01, \tag{50}$$

where the constant hidden in $O(\cdot)$ only depends on the choice of $\alpha$. Since $\log(1/\alpha) \leq 4\log(d)$, the constant hidden in $O(\cdot)$ is absolute. Therefore, the last inequality holds with sufficiently small $c_v$, which does not depend on other parameters. $\qquad\square$

### A.3 Useful Lemmas

In this part we bound some quantities that we frequently encounter as the error terms. These lemmas will simplify our proofs in later parts.

The next lemma helps to bound the "interaction error" arose from the non-orthogonality of $\mathbf{V}^\star$ and $\mathbf{U}^\star$.

**Lemma 8.** *Let $\mathbf{R}_t, \mathbf{Q}_t, \mathbf{E}_t$ and $\epsilon_1$ be defined as above. We have*

$$\left\|\mathbf{U}_t^\top \mathbf{U}_t - \left(\mathbf{R}_t \mathbf{R}_t^\top + \mathbf{Q}_t \mathbf{Q}_t^\top + \mathbf{E}_t^\top \mathbf{E}_t\right)\right\| \leq 6\epsilon_1 \|\mathbf{U}_t\|^2 \tag{51}$$

*Proof.* From the definition of $\mathbf{R}_t, \mathbf{Q}_t, \mathbf{E}_t$, we have

$$\mathbf{U}_t^\top \mathbf{U}_t = \left(\mathbf{U}^\star \mathbf{R}_t^\top + \mathbf{V}^\star \mathbf{Q}_t^\top + \mathbf{E}_t\right)^\top \left(\mathbf{U}^\star \mathbf{R}_t^\top + \mathbf{V}^\star \mathbf{Q}_t^\top + \mathbf{E}_t\right)$$
$$= \mathbf{R}_t \mathbf{R}_t^\top + \mathbf{Q}_t \mathbf{Q}_t^\top + \mathbf{E}_t^\top \mathbf{E}_t + \mathbf{U}_t^\top \left(\mathrm{Id}_{\mathbf{U}^\star} \mathrm{Id}_{\mathrm{res}} + \mathrm{Id}_{\mathbf{V}^\star} \mathrm{Id}_{\mathrm{res}}\right.$$
$$\left. + \mathrm{Id}_{\mathrm{res}} \mathrm{Id}_{\mathbf{U}^\star} + \mathrm{Id}_{\mathrm{res}} \mathrm{Id}_{\mathbf{V}^\star} + \mathrm{Id}_{\mathbf{U}^\star} \mathrm{Id}_{\mathbf{V}^\star} + \mathrm{Id}_{\mathbf{V}^\star} \mathrm{Id}_{\mathbf{U}^\star}\right) \mathbf{U}_t. \tag{52}$$

Note that

$$\left\|\mathrm{Id}_{\mathbf{U}^\star} \mathrm{Id}_{\mathbf{V}^\star}\right\| = \left\|\mathbf{U}^\star \mathbf{U}^{\star\top} \mathbf{V}^\star \mathbf{V}^{\star\top}\right\| \leq \epsilon_1 \tag{53}$$

and

$$\left\|\mathrm{Id}_{\mathrm{res}} \mathrm{Id}_{\mathbf{U}^\star}\right\| = \left\|(\mathbf{I} - \mathrm{Id}_{\mathbf{U}^\star} - \mathrm{Id}_{\mathbf{V}^\star}) \mathrm{Id}_{\mathbf{U}^\star}\right\| = \left\|-\mathrm{Id}_{\mathbf{U}^\star} \mathrm{Id}_{\mathbf{V}^\star}\right\| \leq \epsilon_1. \tag{54}$$

Similarly, for the other terms, we can prove that all the six terms in the bracket in the last line of 52 have operator norm $\leq \epsilon_1$. This completes the proof. $\qquad\square$

The next lemma helps to bound the RIP error in the dynamic of $\mathbf{U}_t$.

**Lemma 9** (Upper Bound for $\mathsf{E}_t$). *Under the assumption of Theorem 2, if $\|\mathbf{E}_t\|, \|\mathbf{Q}_t\|, \|\mathbf{R}_t\| < 1.1$ and $\|\mathbf{E}_t\|_F^2 < 1$, we have that:*

$$\left\|\mathbf{U}_t^\top \mathsf{E}_t \circ \left(\mathbf{U}_t \mathbf{U}_t^\top - \mathbf{U}^\star \mathbf{U}^{\star\top} - \mathbf{V}^\star \boldsymbol{\Sigma}_t \mathbf{V}^{\star\top}\right)\right\| \leq 2M_1 \delta \sqrt{r_1 + r_2} \|\mathbf{U}_t\|. \tag{55}$$

*Proof.*

$$\left\| \mathsf{E}_t \circ \left( \mathbf{U}_t \mathbf{U}_t^\top - \mathbf{U}^\star \mathbf{U}^{\star\top} - \mathbf{V}^\star \boldsymbol{\Sigma}_t \mathbf{V}^{\star\top} \right) \right\|$$

$$\overset{(a)}{\le} \left\| \mathsf{E}_t \circ \left( \mathbf{V}^\star \boldsymbol{\Sigma}_t \mathbf{V}^{\star\top} \right) \right\| + \left\| \mathsf{E}_t \circ \left( \mathbf{E}_t \mathbf{E}_t^\top \right) \right\| + \left\| \mathsf{E}_t \circ \left( \mathbf{U}_t \mathbf{U}_t^\top - \mathbf{U}^\star \mathbf{U}^{\star\top} - \mathbf{E}_t \mathbf{E}_t^\top \right) \right\|$$

$$\overset{(b)}{\le} \delta \Big( \|\boldsymbol{\Sigma}_t\|_F + \left\| \mathbf{E}_t \mathbf{E}_t^\top \right\|_* \tag{56}$$
$$+ \left\| \mathbf{R}_t^\top \mathbf{R}_t - \mathbf{I} \right\|_F + \left\| \mathbf{Q}_t^\top \mathbf{Q}_t \right\|_F + 2\|\mathbf{E}_t\|(\|\mathbf{R}_t\|_F + \|\mathbf{Q}_t\|_F) + 2\left\| \mathbf{Q}_t^\top \mathbf{R}_t \right\|_F \Big)$$
$$\le \delta(\sqrt{r_2}M_1 + 1 + 3\sqrt{r_1} + 4\sqrt{r_2} + 8(\sqrt{r_1} + \sqrt{r_2}) + 8\sqrt{r_1})$$
$$\le 2M_1 \delta \sqrt{r_1 + r_2},$$

where in $(a)$ we use the linearity of $\mathsf{E}_t \circ (\cdot)$ and the triangle inequality. In $(b)$ we use Lemma 2 for the first term, Lemma 4 for the second term, and the expansion:

$$\mathbf{U}_t \mathbf{U}_t^\top - \mathbf{U}^\star \mathbf{U}^{\star\top} - \mathbf{E}_t \mathbf{E}_t^\top = \mathbf{U}^\star(\mathbf{R}_t^\top \mathbf{R}_t - \mathbf{I})\mathbf{U}^{\star\top} + \mathbf{V}^\star \mathbf{Q}_t^\top \mathbf{Q}_t \mathbf{V}^{\star\top}$$
$$+ \mathbf{E}_t(\mathbf{R}_t \mathbf{U}^{\star\top} + \mathbf{Q}_t \mathbf{V}^{\star\top}) + (\mathbf{V}^\star \mathbf{Q}_t^\top + \mathbf{U}^\star \mathbf{R}_t^\top)\mathbf{E}_t^\top \tag{57}$$
$$+ \mathbf{V}^\star \mathbf{Q}_t^\top \mathbf{R}_t \mathbf{U}^{\star\top} + \mathbf{U}^\star \mathbf{R}_t^\top \mathbf{Q}_t \mathbf{V}^{\star\top}.$$

for the third term which shows that $\mathbf{U}_t \mathbf{U}_t^\top - \mathbf{U}^\star \mathbf{U}^{\star\top} - \mathbf{E}_t \mathbf{E}_t^\top$ has rank no more than $2(r_1 + r_2)$. Hence we can conclude that:

$$\left\| \mathbf{U}_t^\top \mathsf{E}_t \circ \left( \mathbf{U}_t \mathbf{U}_t^\top - \mathbf{U}^\star \mathbf{U}^{\star\top} - \mathbf{V}^\star \boldsymbol{\Sigma}_t \mathbf{V}^{\star\top} \right) \right\| \le 2M_1 \delta \sqrt{r_1 + r_2} \cdot \|\mathbf{U}_t\|. \tag{58}$$

$\square$

The following lemma tells how to bound the interaction error and RIP error using the auxiliary sequences $\mathrm{R}_t$ and $\mathrm{L}_t$ we have already defined:

**Lemma 10** (Bound Using calibration Line). *Under the assumptions of Theorem 2, if $\|\mathbf{E}_t\| \le \|\mathbf{R}_t\| \le \min\{4\mathrm{R}_t, 1.1\}$ and $\|\mathbf{Q}_t\| \le \sqrt{r_2}p^{-1.5}r_2^{1.5} \cdot \mathrm{L}_t$, we have*

$$\left( M_1\epsilon_1 + 2M_1\delta\sqrt{r_1 + r_2} \right) \|\mathbf{U}_t\| \le \mathrm{L}_t \wedge \frac{5}{576} \log^{-1}\left( \frac{1}{\alpha} \right) \mathrm{R}_t. \tag{59}$$

*Proof.* From triangle inequality and the condition of this lemma ($2\epsilon_1 \le \delta$), we have that:

$$\left( M_1\epsilon_1 + 2M_1\delta\sqrt{r_1 + r_2} \right) \|\mathbf{U}_t\| \le \frac{5}{2}M_1\delta\sqrt{r_1 + r_2}(\|\mathbf{R}_t\| + \|\mathbf{Q}_t\| + \|\mathbf{E}_t\|)$$
$$\le \frac{5}{2}M_1\delta\sqrt{r_1 + r_2}(8\mathrm{R}_t + \sqrt{r_2}p^{-1.5}r_2^{1.5} \cdot \mathrm{L}_t). \tag{60}$$

Then it suffices to check:

$$\begin{cases} 20M_1\delta\sqrt{r_1 + r_2}\mathrm{R}_t \overset{(a)}{\le} \frac{1}{2}\mathrm{L}_t; \\ \frac{2}{5}M_1\delta\sqrt{r_2}p^{-1.5}r_2^{1.5}\sqrt{r_1 + r_2}\mathrm{L}_t \overset{(b)}{\le} \frac{1}{2}\mathrm{L}_t; \\ 20M_1\delta\sqrt{r_1 + r_2}\mathrm{R}_t \overset{(c)}{\le} \frac{5}{1152}\log^{-1}\left(\frac{1}{\alpha}\right)\mathrm{R}_t; \\ \frac{2}{5}M_1\delta\sqrt{r_2}p^{-1.5}r_2^{1.5}\sqrt{r_1 + r_2}\mathrm{L}_t \overset{(d)}{\le} \frac{5}{1152}\log^{-1}\left(\frac{1}{\alpha}\right)\mathrm{R}_t, \end{cases}$$

where $(a)$ is from the definition of $\mathrm{L}_t$, $(b)$ and $(c)$ are from the assumption on $\delta$ in Theorem 2, and $(d)$ is from the assumption on $\delta$ (the absolute constant $c$ in the condition for $\delta$) and the fact that $\mathrm{L}_t \le \mathrm{R}_t$. Hence the proof is completed. $\square$

## A.4 Bounds of $\mathbf{Q}_t$

For evaluating the magnitude of $\|\mathbf{Q}_t\|$, we consider its columns. We denote each column of $\mathbf{Q}_t$ as $\mathbf{q}_i^{(t)}$ for $i = 1, 2, \ldots, r_2$. And use $q_i^t$ we defined above to upper bound them. Once we provide a uniform bound for all $\mathbf{q}_i^{(t)}$, we can also bound $\|\mathbf{Q}_t\|$.

**Lemma 11.** *Under the assumption of Theorem 2, under the event of $q_i^t < p^{-1.5} r_2^{1.5} \cdot L_t$ with $p \geq \epsilon_2^{2/3}$ for all $i = 1, 2, \ldots, r_2$ and $t = 0, 1, \ldots, T$, if $\|\mathbf{E}_t\| \leq \|\mathbf{R}_t\| \leq 1.1$, $\|\mathbf{E}_t\|_F^2 < 1$ and $\|\mathbf{q}_j^t\| \leq q_j^t$ for all $j = 1, \ldots, r_2$, then we have*

$$\|\mathbf{q}_i^{(t+1)}\| \leq q_i^{t+1} < p^{-1.5} r_2^{1.5} L_{t+1} \tag{61}$$

*for all $i = 1, 2, \ldots, r_2$ .*

*Proof.* From the dynamic of $\mathbf{Q}_t$:

$$\mathbf{Q}_{t+1} = \mathbf{Q}_t - \eta \mathbf{U}_t^\top \mathbf{U}_t \mathbf{Q}_t + \eta \mathbf{Q}_t \boldsymbol{\Sigma} - \eta \left[ \left( \epsilon_1 + 2M_1 \delta \sqrt{r_1 + r_2} \right) \|\mathbf{U}_t\| \right],$$

we can see that for each column $\mathbf{q}_i^{(t)}$ of $\mathbf{Q}_t$:

$$\|\mathbf{q}_i^{(t+1)}\| \leq \left\| \left( \mathbf{I} - \eta \mathbf{U}_t^\top \mathbf{U}_t + \eta \boldsymbol{\Sigma}_{ii}^{(e_t)} \mathbf{I} \right) \mathbf{q}_i^{(t)} \right\| + \eta \sum_{j \neq i} |\boldsymbol{\Sigma}_{ji}^{(e_t)}| \|\mathbf{q}_j^{(t)}\| + \eta \left( \epsilon_1 + 2M_1 \delta \sqrt{r_1 + r_2} \right) \|\mathbf{U}_t\|$$

$$\leq (1 + \eta \boldsymbol{\Sigma}_{ii}^{(e_t)}) \|\mathbf{q}_i^{(t)}\|_2 + \eta \sum_{j \neq i} |\boldsymbol{\Sigma}_{ji}^{(e_t)}| \|\mathbf{q}_j^{(t)}\| + \eta L_t.$$

where we use Lemma 10. For the second term we have:

$$\eta \sum_{j \neq i} |\boldsymbol{\Sigma}_{ji}^{(e_t)}| \|\mathbf{q}_j^{(t)}\| \overset{(a)}{\leq} \eta \frac{c_o}{r^2 M_2^{1.5}} p^{-1.5} r_2^{1.5} L_t \overset{(b)}{\leq} \eta L_t (c_o c_v^{-1.5} r^{-0.5} \log^{1.5} d) \overset{(c)}{\leq} 1, \tag{62}$$

where in $(a)$ we use Assumption 1 (c) and induction hypothesis that $\|\mathbf{q}_j^{(t)}\| < p^{-1.5} r_2^{1.5} \cdot L_t$, in $(b)$ we use the definition of $p$ (Definition 4), and in $(c)$ we use Assumption 1 (a) and Assumption 2 with sufficiently small $c_o$ (which depends solely on another universal constant $c_v$). Hence we have

$$\|\mathbf{q}_i^{(t+1)}\| \leq (1 + \eta \boldsymbol{\Sigma}_{ii}^{(e_t)}) \|\mathbf{q}_i^{(t)}\|_2 + 2\eta L_t. \tag{63}$$

There are two probable cases:

$$\begin{cases} \text{If } \|\mathbf{q}_i^{(t)}\| \leq L_t, \text{ then } \|\mathbf{q}_i^{(t+1)}\| \leq (1 + \eta \boldsymbol{\Sigma}_{ii}^t + 2\eta) L_t \leq (1 + \eta \boldsymbol{\Sigma}_{ii}^t + 2\eta) q_i^t \leq q_i^{t+1}; \\ \text{If } \|\mathbf{q}_i^{(t)}\| > L_t, \text{ then } \|\mathbf{q}_i^{(t+1)}\| \leq (1 + \eta \boldsymbol{\Sigma}_{ii}^t + 2\eta) \|\mathbf{q}_i^t\| \leq (1 + \eta \boldsymbol{\Sigma}_{ii}^t + 2\eta) q_i^t \leq q_i^{t+1}. \end{cases}$$

Both lead to the results we desire. $\square$

With the above lemma, we can also give a bound for $\|\mathbf{Q}_t\|$:

**Corollary 1.** *Under the condition of Lemma 11, we have that*

$$\|\mathbf{Q}_{t+1}\| \leq \|\mathbf{Q}_{t+1}\|_F \leq \sqrt{\sum_{i=1}^{r_2} (q_i^{t+1})^2} < p^{-1.5} r_2^{1.5} \sqrt{r_2} L_{t+1}. \tag{64}$$

## A.5 Bounds of $\mathbf{E}_t$

In this section, we bound the increments of both the operator norm and the Frobenius norm of $\mathbf{E}_t$. The next lemma provide an upper bound for $\|\mathbf{E}_t\|$.

**Lemma 12** (Increment of Spectral Norm of $\mathbf{E}_t$). *Under the assumption of Lemma 11, we have*

$$\|\mathbf{E}_{t+1}\| \leq \|\mathbf{E}_t\| + \eta L_t. \tag{65}$$

*Proof.* From the dynamic of $\mathbf{E}_t$ (24), we can derive that

$$\|\mathbf{E}_{t+1}\| \leq \left\|\mathbf{E}_t\left(\mathbf{I} - \eta\mathbf{U}_t^\top\mathbf{U}_t\right)\right\| + \eta\left(\left\|\mathrm{Id}_{\mathrm{res}}\left(\mathbf{U}^\star\mathbf{U}^{\star\top} + \mathbf{V}^\star\boldsymbol{\Sigma}_t\mathbf{V}^{\star\top}\right)\right\| + \|\mathrm{Id}_{\mathrm{res}}\mathsf{E}_t\|\right)\|\mathbf{U}_t\|$$

$$\overset{(a)}{\leq} \|\mathbf{E}_t\| + \eta\left(\left(\epsilon_1 + M_1 + 2M_1\delta\sqrt{r_1 + r_2}\right)\|\mathbf{U}_t\|\right.$$

$$\overset{(b)}{\leq} \|\mathbf{E}_t\| + \eta 5M_1\delta\sqrt{r_1 + r_2}\left(\|\mathbf{R}_t\| + \|\mathbf{Q}_t\|\right)$$

$$\overset{(c)}{\leq} \|\mathbf{E}_t\| + \eta\mathsf{L}_t,$$

where $(a)$ is from Lemma 9 and the fact that $\|\mathrm{Id}_{\mathrm{res}}\mathbf{U}_t\|, \|\mathrm{Id}_{\mathrm{res}}\mathbf{U}_t\| \leq \epsilon_1$. $(b)$ and $(c)$ are derived similarly as Lemma 10. $\qquad\square$

The next lemma bounds the F-norm of error component $\mathbf{E}_t$.

**Lemma 13** (Increment of the F-norm of Error Dynamic). *Under the assumption of Lemma 11, and we further assume that $\|\mathbf{E}_t\| \lesssim \delta M_1\sqrt{r_1 + r_2}\log(1/\alpha)$, then the Frobenius norm of $\mathbf{E}_{t+1}$ can be bounded by*

$$\|\mathbf{E}_{t+1}\|_F^2 \leq (1 + O(\eta\delta M_1\sqrt{r_1 + r_2}))\|\mathbf{E}_t\|_F^2 + \eta O(\delta^2 M_1^2(r_1 + r_2)^{1.5}\log(1/\alpha)), \qquad (66)$$

*which immediately implies,*

$$\|\mathbf{E}_t\|_F^2 \lesssim \left((1 + O(\eta\delta M_1\sqrt{r_1 + r_2}))^t - 1\right)\delta M_1(r_1 + r_2)\log(1/\alpha)$$
$$\lesssim t\eta\delta^2 M_1^2(r_1 + r_2)^{1.5}\log(1/\alpha). \qquad (67)$$

*Proof.* We expand $\|\mathbf{E}_{t+1}\|_F^2$ from the dynamic of $\mathbf{E}_t$ (24):

$$\|\mathbf{E}_{t+1}\|_F^2 = \left\|\mathbf{E}_t\left(\mathbf{I} - \eta\mathbf{U}_t^\top\mathbf{U}_t\right) + \eta\mathrm{Id}_{\mathrm{res}}\left(\mathbf{U}^\star\mathbf{U}^{\star\top} + \mathbf{V}^\star\boldsymbol{\Sigma}_t\mathbf{V}^{\star\top}\right)\mathbf{U}_t - \eta\mathrm{Id}_{\mathrm{res}}\mathsf{E}_t\mathbf{U}_t\right\|_F^2$$

$$= \|\mathbf{E}_t(\mathbf{I} - \eta\mathbf{U}_t^\top\mathbf{U}_t)\|_F^2 + \eta^2\|\mathrm{Id}_{\mathrm{res}}\mathsf{E}_t\mathbf{U}_t\|_F^2$$

$$\quad - 2\eta\left\langle\mathbf{E}_t(\mathbf{I} - \eta\mathbf{U}_t^\top\mathbf{U}_t), \mathrm{Id}_{\mathrm{res}}\mathsf{E}_t\mathbf{U}_t\right\rangle$$

$$\quad + \left\|\eta\mathrm{Id}_{\mathrm{res}}\left(\mathbf{U}^\star\mathbf{U}^{\star\top} + \mathbf{V}^\star\boldsymbol{\Sigma}_t\mathbf{V}^{\star\top}\right)\mathbf{U}_t\right\|_F^2 \qquad (68)$$

$$\quad + \left\langle\mathbf{E}_t\left(\mathbf{I} - \eta\mathbf{U}_t^\top\mathbf{U}_t\right), \eta\mathrm{Id}_{\mathrm{res}}\left(\mathbf{U}^\star\mathbf{U}^{\star\top} + \mathbf{V}^\star\boldsymbol{\Sigma}_t\mathbf{V}^{\star\top}\right)\mathbf{U}_t\right\rangle$$

$$\quad + \left\langle\eta\mathrm{Id}_{\mathrm{res}}\left(\mathbf{U}^\star\mathbf{U}^{\star\top} + \mathbf{V}^\star\boldsymbol{\Sigma}_t\mathbf{V}^{\star\top}\right)\mathbf{U}_t, \eta\mathrm{Id}_{\mathrm{res}}\mathsf{E}_t\mathbf{U}_t\right\rangle$$

$$\overset{\mathrm{def}}{=} (1) + (2) + (3) + (4) + (5) + (6).$$

Now we bound the six parts separately. For the first part, since $0 \preceq \eta\mathbf{U}_t^\top\mathbf{U}_t \preceq \mathbf{I}$, we have

$$\|\mathbf{E}_t(\mathbf{I} - \eta\mathbf{U}_t^\top\mathbf{U}_t)\|_F^2 \leq \|\mathbf{E}_t\|_F^2. \qquad (69)$$

For the second part:

$$(2) \leq \eta^2\left\langle\mathsf{E}_t, \mathrm{Id}_{\mathrm{res}}^2\mathsf{E}_t\mathbf{U}_t\mathbf{U}_t^\top\right\rangle$$

$$\overset{(a)}{\leq} \eta^2\delta\left(\|\mathbf{U}_t\mathbf{U}_t^\top - \mathbf{U}^\star\mathbf{U}^{\star\top} - \mathbf{E}_t\mathbf{E}_t^\top\|_F + \|\mathbf{E}_t\mathbf{E}_t^\top\|_* + \|\mathbf{V}^\star\boldsymbol{\Sigma}_t\mathbf{V}^{\star\top}\|_F\right)$$

$$\left(\left\|\mathsf{E}_t(\mathbf{U}_t\mathbf{U}_t^\top - \mathbf{E}_t\mathbf{E}_t^\top)\right\|_F + \left\|\mathsf{E}_t\mathbf{E}_t\mathbf{E}_t^\top\right\|_*\right)$$

$$\leq \eta^2\delta\left(\|\mathbf{U}_t\mathbf{U}_t^\top - \mathbf{U}^\star\mathbf{U}^{\star\top} - \mathbf{E}_t\mathbf{E}_t^\top\|_F + \|\mathbf{E}_t\mathbf{E}_t^\top\|_* + \|\mathbf{V}^\star\boldsymbol{\Sigma}_t\mathbf{V}^{\star\top}\|_F\right) \qquad (70)$$

$$\|\mathsf{E}_t\|\left(\left\|\mathbf{U}_t\mathbf{U}_t^\top - \mathbf{E}_t\mathbf{E}_t^\top\right\|_F + \|\mathbf{E}_t\mathbf{E}_t^\top\|_*\right)$$

$$\overset{(b)}{\lesssim} \eta^2\delta M_1\sqrt{r_1 + r_2} \cdot \delta M_1\sqrt{r_1 + r_2} \cdot \left(O(\sqrt{r_1 + r_2}) + \|\mathbf{E}_t\|_F^2\right)$$

$$\lesssim \eta^2\delta^2 M_1^2(r_1 + r_2)^{1.5}.$$

In $(a)$ we use similar technique as in Lemma 9 to divide $\mathbf{U}_t\mathbf{U}_t^\top - \mathbf{U}^\star\mathbf{U}^{\star\top} - \mathbf{V}^\star\mathbf{\Sigma}_t\mathbf{V}^{\star\top}$ into three parts so that we can use Lemma 1 and Lemma 3. In $(b)$ we use Lemma 9 to bound the first two terms and (57) to bound the third term with the assumption that $\|\mathbf{R}_t\|, \|\mathbf{Q}_t\|, \|\mathbf{E}_t\| < 2$.

For the third part:

$$
\begin{aligned}
(3) &\overset{(a)}{=} -2\eta\left\langle \mathsf{E}_t, \mathrm{Id}_{\mathrm{res}}\mathbf{E}_t(\mathbf{I}-\eta\mathbf{U}_t^\top\mathbf{U}_t)\mathbf{U}_t^\top\right\rangle \\
&= -2\eta\left\langle \mathsf{E}_t, \mathrm{Id}_{\mathrm{res}}\mathbf{E}_t(\mathbf{I}-\eta\mathbf{U}_t^\top\mathbf{U}_t)(\mathbf{E}_t^\top + \mathbf{R}_t\mathbf{U}^{\star\top} + \mathbf{Q}_t\mathbf{V}^{\star\top})\right\rangle \\
&\overset{(b)}{\le} 2\eta\delta\left( \|\mathbf{U}_t\mathbf{U}_t^\top - \mathbf{U}^\star\mathbf{U}^{\star\top} - \mathbf{E}_t\mathbf{E}_t^\top\|_F + \|\mathbf{E}_t\mathbf{E}_t^\top\|_* + \|\mathbf{\Sigma}_t\|_F \right) \\
&\quad \cdot \left( \|\mathbf{E}_t(\mathbf{I}-\eta\mathbf{U}_t^\top\mathbf{U}_t)\mathbf{E}_t^\top\|_* + \|\mathbf{E}_t(\mathbf{I}-\eta\mathbf{U}_t^\top\mathbf{U}_t)\mathbf{R}_t\|_F + \|\mathbf{E}_t(\mathbf{I}-\eta\mathbf{U}_t^\top\mathbf{U}_t)\mathbf{Q}_t\|_F \right) \\
&\overset{(c)}{\le} 2\eta\delta O(M_1\sqrt{r_1+r_2})\left( \|\mathbf{E}_t\|_F^2 + \|\mathbf{E}_t\|\|\mathbf{R}_t\|_F + \|\mathbf{E}_t\|\|\mathbf{Q}_t\|_F \right) \\
&\le 2\eta\delta O(M_1\sqrt{r_1+r_2})\left( \|\mathbf{E}_t\|_F^2 + (2\sqrt{r_1}+2\sqrt{r_2})\|\mathbf{E}_t\| \right) \\
&\lesssim \eta\delta M_1\sqrt{r_1+r_2}\|\mathbf{E}_t\|_F^2 + \eta\delta^2 M_1^2(r_1+r_2)^{1.5}\log(1/\alpha),
\end{aligned}
$$
$$(71)$$

where in $(a)$ we use the fact that $\langle \mathbf{A},\mathbf{BCD}\rangle = \langle \mathbf{B}^\top\mathbf{AD}^\top,\mathbf{C}\rangle$. In $(b)$ we separate $\mathsf{E}_t$ as in (70). In $(c)$, for the first term we use the upper bound for $\mathsf{E}_t$ appeared in Lemma 9, for the second term we use $\|\mathbf{AB}\|_F \le \|\mathbf{A}\|\|\mathbf{B}\|_F$ (and similarly for nuclear norm) and $\|\mathbf{I}-\eta\mathbf{U}_t^\top\mathbf{U}_t\| \le 1$.

For the fourth part:

$$(4) \le 2\eta^2\epsilon_1^2\|\mathbf{R}_t\|_F^2 + 2\eta^2\epsilon_1^2\|\mathbf{Q}_t\|_F^2 \le 8\eta^2\epsilon_1^2(r_1+r_2), \qquad (72)$$

where the first inequality is from Cauchy's inequality and the fact that $\|\mathrm{Id}_{\mathrm{res}}\mathbf{U}^\star\|, \|\mathrm{Id}_{\mathrm{res}}\mathbf{V}^\star\| \le \epsilon_1$.

For the fifth part:

$$
\begin{aligned}
(5) &\le \left\|\mathbf{E}_t\left(\mathbf{I}-\eta\mathbf{U}_t^\top\mathbf{U}_t\right)\right\| \cdot \left\|\eta\mathrm{Id}_{\mathrm{res}}\left(\mathbf{U}^\star\mathbf{U}^{\star\top} + \mathbf{V}^\star\mathbf{\Sigma}_t\mathbf{V}^{\star\top}\right)\mathbf{U}_t\right\|_* \\
&\lesssim \|\mathbf{E}_t\|\eta\epsilon_1 M_1(r_1+r_2) \\
&\lesssim \eta\left(\delta M_1\sqrt{r_1+r_2}\log(1/\alpha)\right)\epsilon_1 M_1(r_1+r_2) \\
&\le \eta\delta^2 M_1^2(r_1+r_2)^{1.5}\log(1/\alpha),
\end{aligned}
\qquad (73)
$$

where the first inequality is from the norm inequality $\langle \mathbf{X},\mathbf{Y}\rangle \le \|\mathbf{X}\|_*\|\mathbf{Y}\|$ and in the last inequality we use the fact that $\epsilon_1 < \delta$.

For the sixth part:

$$
\begin{aligned}
(6) &= \eta^2\left\langle \mathsf{E}_t, \mathrm{Id}_{\mathrm{res}}^2\left(\mathbf{U}^\star\mathbf{U}^{\star\top} + \mathbf{V}^\star\mathbf{\Sigma}_t\mathbf{V}^{\star\top}\right)\mathbf{U}_t\mathbf{U}_t^\top\right\rangle \\
&\overset{(a)}{\le} \eta^2\|\mathsf{E}_t\| \cdot \|\mathrm{Id}_{\mathrm{res}}^2\left(\mathbf{U}^\star\mathbf{U}^{\star\top} + \mathbf{V}^\star\mathbf{\Sigma}_t\mathbf{V}^{\star\top}\right)\mathbf{U}_t\mathbf{U}_t^\top\|_* \\
&\le \eta^2\|\mathsf{E}_t\| \cdot \|\mathbf{U}_t\|^2 \cdot \|\mathrm{Id}_{\mathrm{res}}^2\left(\mathbf{U}^\star\mathbf{U}^{\star\top} + \mathbf{V}^\star\mathbf{\Sigma}_t\mathbf{V}^{\star\top}\right)\|_* \\
&\overset{(b)}{\lesssim} \eta^2\delta M_1\sqrt{r_1+r_2} \cdot \|\mathbf{U}_t\|^2\epsilon_1(r_1+M_1 r_2) \\
&\lesssim \eta^2\delta M_1\epsilon_1 M_1(r_1+r_2)^{1.5} \\
&\lesssim \eta^2\delta^2 M_1^2(r_1+r_2)^{1.5}.
\end{aligned}
\qquad (74)
$$

In $(a)$ we use the norm inequality $\langle \mathbf{X},\mathbf{Y}\rangle \le \|\mathbf{X}\|_*\|\mathbf{Y}\|$ and in $(b)$ we use $\|\mathrm{Id}_{\mathrm{res}}\| \le 1$ and $\|\mathrm{Id}_{\mathrm{res}}\mathbf{U}_t\|, \|\mathrm{Id}_{\mathrm{res}}\mathbf{U}_t\| \le \epsilon_1$.

Now combining Equation (69)-(74), along with the fact that $\|\mathbf{E}_0\|_F^2 \le d\alpha^2 < d^{-1} \ll \delta^2 r^{1.5}$, we can derive the result we desire.

$\square$

### A.6 Analysis for Phase 1

In this section, we give a rigorous analysis for phase 1:

**Theorem 4** (Phase 1 analysis). *Under the assumptions of Theorem 2. During the first $T_1 = O(\frac{1}{\eta}\log(\frac{1}{\alpha}))$ steps, with probability at least $0.995$, the following holds for any $t \in [0, T_1]$, that*

- $\sigma_j(\mathbf{R}_{t+1}) > (1 + \eta/3)\sigma_j(\mathbf{R}_t)$ *for all* $j \in [r_1]$;

- $\|\mathbf{Q}_t\|_F \leq p^{-1.5}r_2^{1.5}\sqrt{r_2} \cdot \mathrm{L}_t \leq \delta^\star < 0.01$, *where* $\mathrm{L}_t$ *is formally defined in* (38).

- $\|\mathbf{E}_t\| \lesssim \delta\sqrt{r_1 + r_2} \leq \|\mathbf{R}_t\|$ *and* $\|\mathbf{E}_t\|_F^2 \lesssim \delta^2 M_1^2(r_1 + r_2)^{1.5}\log(1/\alpha)^2$.

*Finally, we have* $\sigma_1(\mathbf{R}_{T_1}), \sigma_{r_1}(\mathbf{R}_{T_1}) \in \left(\frac{1}{4}, \frac{7}{18}\right)$.

In the below contexts, unless otherwise specified, we abbreviate the largest and smallest singular values of $\mathbf{R}_t$ as $\sigma_1^{(t)}$ and $\sigma_{r_1}^{(t)}$.

The next lemma tells that, if $\|\mathbf{Q}_t\|$ and $\|\mathbf{E}_t\|$ are both small, then $\mathbf{R}_t$ increases steadily and the deviation between its singular values is small.

**Lemma 14** (Dynamic of Singular Values of $\mathbf{R}_t$ in Phase 1). *For some $t \leq T_1 - 1$, under the assumptions of Lemma 11, if $\|\mathbf{E}_t\|, \|\mathbf{Q}_t\| < \frac{1}{96}\log^{-1}(1/\alpha)$ and $\underline{\mathrm{R}}_t \leq \sigma_{r_1}^{(t)} \leq \sigma_1^{(t)} \leq \overline{\mathrm{R}}_t$, then*

$$\underline{\mathrm{R}}_{t+1} \leq \sigma_{r_1}^{(t+1)} \leq \sigma_1^{(t+1)} \leq \overline{\mathrm{R}}_{t+1}. \tag{75}$$

*and*

$$\begin{aligned}
\sigma_1^{(t+1)} &\geq (1 + \eta/3)\sigma_1^{(t)} \\
\sigma_{r_1}^{(t+1)} &\geq (1 + \eta/3)\sigma_{r_1}^{(t)}
\end{aligned} \tag{76}$$

*Proof.* From the dynamic of $\mathbf{R}_t$:

$$\begin{aligned}
\mathbf{R}_{t+1} &= (\mathbf{I} - \eta\mathbf{U}_t^\top\mathbf{U}_t + \eta\mathbf{I})\mathbf{R}_t + \eta\mathbf{U}_t^\top\mathbf{V}^\star\boldsymbol{\Sigma}_t\mathbf{V}^{\star\top}\mathbf{U}^\star - \eta\mathbf{U}_t^\top\mathsf{E}_t\mathbf{U}^\star \\
&= (\mathbf{I} - \eta\mathbf{R}_t\mathbf{R}_t^\top + \eta\mathbf{I})\mathbf{R}_t - \eta(\mathbf{Q}_t\mathbf{Q}_t^\top + \mathbf{E}_t^\top\mathbf{E}_t)\mathbf{R}_t + \eta\mathbf{U}_t^\top\left[(M_1\epsilon_1 + 2M_1\delta\sqrt{r + r_2})\right].
\end{aligned}$$

We use $\sigma_1^{(t)}$ and $\sigma_{r_1}^{(t)}$ to denote the largest/smallest singular value of $\mathbf{R}_t$. To control the dynamic of $\sigma_1^{(t)}$ and $\sigma_{r_1}^{(t)}$, we need to bound the magnitude of the error term, that is

$$\begin{aligned}
&\left\|\eta(\mathbf{Q}_t\mathbf{Q}_t^\top + \mathbf{E}_t^\top\mathbf{E}_t)\mathbf{R}_t + \eta\mathbf{U}_t^\top[2.5\delta M_1\sqrt{r_1 + r_2}]\right\| \\
&\leq \eta\left(\|\mathbf{Q}_t\|^2 + \|\mathbf{E}_t\|^2 + \frac{1}{32}\log^{-1}(1/\alpha)\right)\sigma_1^{(t)} \\
&\leq \eta(\frac{1}{96} + \frac{1}{96} + \frac{1}{96})\log^{-1}(1/\alpha)\,\sigma_1^{(t)} \\
&\leq \frac{\eta}{32}\log^{-1}(1/\alpha)\,\sigma_1^{(t)},
\end{aligned} \tag{77}$$

where in the first inequality we use the assumption for $\|\mathbf{Q}_t\|$ and $\|\mathbf{E}_t\|$ and $\delta$. Therefore, from Weyl's inequality, we have that

$$\begin{cases}
\sigma_1^{(t+1)} \leq (1 - \eta\sigma_1^{(t)2} + \eta)\sigma_1^{(t)} + \frac{\eta}{32}\log(1/\alpha)^{-1}\sigma_1^{(t)}; \\
\sigma_{r_1}^{(t+1)} \geq (1 - \eta\sigma_{r_1}^{(t)2} + \eta)\sigma_{r_1}^{(t)} - \frac{\eta}{32}\log(1/\alpha)^{-1}\sigma_1^{(t)}.
\end{cases} \tag{78}$$

Using the assumption that

$$\sigma_1^{(t)} \leq \overline{\mathrm{R}}_t, \quad \sigma_{r_1}^{(t)} \geq \underline{\mathrm{R}}_t, \quad t = 0, 1, \ldots, T_1. \tag{79}$$

And Lemma 5, we can conclude that

$$(1 - 1/6)\mathrm{R}_{t+1} \leq \underline{\mathrm{R}}_{t+1} \leq \sigma_{r_1}^{(t+1)} \leq \sigma_1^{(t+1)} \leq \overline{\mathrm{R}}_{t+1} \leq (1 + 1/6)\mathrm{R}_{t+1}. \tag{80}$$

For the increasing speed of $\sigma_{r_1}^{(t)}$, note that $\sigma_1^{(t)} < 2\sigma_{r_1}^{(t)}$, therefore

$$\begin{cases}
\sigma_1^{(t+1)} \geq (1 - \frac{1}{4}\eta + \eta - \frac{1}{32}\eta)\sigma_1^{(t)}; \\
\sigma_{r_1}^{(t+1)} \geq (1 - \frac{1}{4}\eta + \eta - \frac{2}{32}\eta)\sigma_{r_1}^{(t)}.
\end{cases} \tag{81}$$

This proves the desired result. $\qquad\square$

Now that the supporting lemmas are prepared, we can begin the proof of Theorem 4

*Proof of Theorem 4.* The initial value of $\mathbf{U}_0$ implies that

$$\|\mathbf{U}_0\| = \|\mathbf{R}_0\| = \|\mathbf{Q}_0\| = \|\mathbf{E}_0\| = \alpha, \quad \|\mathbf{E}_0\|_F^2 \leq \alpha^2 d. \tag{82}$$

Recall that the time $T_1 \leq \frac{5}{\eta} \log(1/\alpha)$ is the first time $\mathrm{R}_t$ enters the region $(1/3 - \eta, 1/3)$. We have that the event of $q_i^t < p^{-1.5} r_2^{1.5} \cdot \mathrm{L}_t$ for all $i = 0, \ldots, r_2, t = 0, \ldots, T_1$ happens with probability over 0.995. In this event, we can use Lemma 11, 12, 13 and 14 to inductively prove:

- For the operator norm of $\mathbf{E}_t$, we have that for all $t \leq T_1$:

$$
\begin{aligned}
\|\mathbf{E}_t\| &\leq \alpha + \eta \sum_{t=0}^{T} \mathrm{L}_t \\
&\leq \alpha(1 + \eta \cdot \frac{240}{\eta} \log\left(\frac{1}{\alpha}\right)) \\
&\quad + 40 M_1 \delta \sqrt{r_1 + r_2} \cdot \frac{\eta}{3} \cdot \left(1 + (1 + \eta/3)^{-1} + (1 + \eta/3)^{-2} + \cdots\right) \\
&\leq 250\alpha \log\left(\frac{1}{\alpha}\right) + 40 M_1 \delta \sqrt{r_1 + r_2}(1 + \eta/3) \\
&\leq 40 M_1 \delta \sqrt{r_1 + r_2} \\
&< \frac{1}{96} \log^{-1}(1/\alpha).
\end{aligned}
\tag{83}
$$

  where in the second inequality we use Lemma 14 that $\|\mathbf{R}_t\|$ increases with rate not less than $(1 + 3/\eta)$.

- For the Frobenius norm of $\mathbf{E}_t$:
$$\|\mathbf{E}_t\|_F^2 \leq T_1 \eta \delta^2 M_1^2 (r_1 + r_2)^{1.5} \log(1/\alpha) \lesssim \delta^2 M_1^2 (r_1 + r_2)^{1.5} \log^2(1/\alpha) < 1 \tag{84}$$

- For $\|\mathbf{Q}_t\|$, we use Corollary 1:
$$p^{-1.5} r_2^{1.5} \sqrt{r_2} \mathrm{L}_{T_1} \lesssim p^{-1.5} r_2^{1.5} \sqrt{r_2} \delta M_1 \sqrt{r_1 + r_2} < \frac{1}{96} \log^{-1}(1/\alpha). \tag{85}$$

- For $\mathbf{R}_t$, we have for $t \leq T_1$:
$$(1 - 1/6)\mathrm{R}_t \leq \underline{\mathrm{R}}_t \leq \sigma_{r_1}^{(t+1)} \leq \sigma_1^{(t+1)} \leq \overline{\mathrm{R}}_t \leq (1 + 1/6)\mathrm{R}_t. \tag{86}$$

- For the condition $\|\mathbf{E}_t\| \leq \|\mathbf{R}_t\|$:
$$\|\mathbf{E}_{t+1}\| - \|\mathbf{E}_t\| \leq \eta \mathrm{L}_t \leq \frac{\eta}{10} \mathrm{R}_t < \frac{\eta}{5} \|\mathbf{R}_t\| < \|\mathbf{R}_{t+1}\| - \|\mathbf{R}_t\|. \tag{87}$$

Hence the proof is completed.

$\square$

## A.7 Analysis for Phase 2

In phase 1, the signal component $\mathbf{R}_t$ grows at a stable speed from $\alpha$ to $O(1)$ while the spurious component $\mathbf{Q}_t$ and the error component $\mathbf{E}_t$ are kept at low levels. In phase 2, we will characterize how $\mathbf{R}_t$ approach 1 and how to continually keep $\mathbf{Q}_t$ and $\mathbf{E}_t$.

**Lemma 15** (Stability of $\mathbf{R}_t$). *If there exists some real number g satisfying*

$$0.01 > g \geq \|\mathbf{Q}_t\|^2 + \|\mathbf{E}_t\|^2 + 4\|\mathbf{U}_t^\top \mathsf{E}_t\| \tag{88}$$

*for all $t = T_1 + 1, \ldots, T_1 + T - 1$, then we have*

$$1 - 5g \leq \sigma_{r_1}^{(t)} \leq \sigma_1^{(t)} \leq 1 + g, \tag{89}$$

*for all $t = T_1 + O(\frac{1}{\eta} \log\left(\frac{1}{g}\right)), \ldots, T_1 + T - 1$*

*Proof.* First we consider the upper bound for $\sigma_1^{(t)}$. Similar to Equation (78), we have

$$\sigma_1^{(t+1)} \leq (1 - \eta\sigma_1^{(t)^2} + \eta + \eta g)\sigma_1^{(t)}. \tag{90}$$

Note that Equation (90) is equivalent to

$$\sqrt{1+g} - \sigma_1^{(t+1)} \geq (\sqrt{1+g} - \sigma_1^{(t)})\left(1 - \eta(\sigma_1^{(t)} + \sqrt{1+g})\sigma_1^{(t)}\right). \tag{91}$$

With $\sigma_1^{(t)}(T_1) < \frac{1}{2}$, one can see that $\sigma_1^{(t)}$ never goes above $\sqrt{1+g} \leq 1 + g$.

Now we consider $\sigma_{r_1}^{(t)}$. After phase 1 we have $\sigma_r^{(T_1)} \geq \frac{5}{6}(1/3 - \eta) > \frac{1}{4}$. If $\sigma_1^{(t)} \leq 5\sigma_{r_1}^{(t)}$, similarly we have:

$$\sigma_{r_1}^{(t+1)} \geq (1 - \eta\sigma_1^{(t)^2} + \eta - 5\eta g)\sigma_{r_1}^{(t)}. \tag{92}$$

Which implies that, if $\sigma_{r_1}^{(t)} < \sqrt{1 - 5g}$,

$$\begin{aligned}
\sqrt{1 - 5g} - \sigma_{r_1}^{(t+1)} &\leq (\sqrt{1 - 5g} - \sigma_{r_1}^{(t)})(1 - \eta(\sigma_{r_1}^{(t)} + \sqrt{1 - 5g})\sigma_{r_1}^{(t)}) \\
&\leq (1 - \frac{1}{4}\eta)(\sqrt{1 - 5g} - \sigma_{r_1}^{(t)})
\end{aligned} \tag{93}$$

Therefore, $\sigma_{r_1}^{(t)}$ will get larger than $\sqrt{1 - 5g} - g^2 \geq 1 - 5g$ at some time $t \leq T_1 + \frac{8}{\eta}\log\left(\frac{1}{g}\right)$. Also from Equation (92) we can see that, $\sigma_{r_1}^{(t)}$ keeps increasing before it gets larger than $\sqrt{1 - 5g}$. And once it surpasses $\sqrt{1 - 5g}$, it never falls below than $\sqrt{1 - 5g}$ again. Therefore, $\sigma_1^{(t)} \leq 5\sigma_{r_1}^{(t)}$ is satisfied and the proof proceeds. $\qquad\square$

Now we can state and prove:

**Theorem 5** (Phase 2 Analysis). *Under the assumptions of Theorem 2. Let $T_2 = T_1 + O(\frac{1}{\eta}\log((r_1 + r_2)/\delta)) \leq O(\frac{1}{\eta}\log(\frac{1}{\alpha}))$. Then with probability at least 0.995, we have*

$$\sigma_1(\mathbf{R}_{T_2}), \sigma_r(\mathbf{R}_{T_2}) \in \left(1 - O(\delta^{\star 2}M_1^2 \vee \delta M_1\sqrt{r_1 + r_2}), 1 + O(\delta^{\star 2}M_1^2 \vee \delta M_1\sqrt{r_1 + r_2})\right). \tag{94}$$

*And for $t = T_1 + 1, \ldots, T_2$, we have*

- $\|\mathbf{Q}_t\|_F \leq p^{-1.5}r_2^{1.5}\sqrt{r_2}\mathrm{L}_t \leq p^{-1.5}r_2^{1.5}\sqrt{r_2}40M_1\delta\sqrt{r_1 + r_2} = 40M_1\delta^\star;$
- $\|\mathbf{E}_t\| \lesssim \delta M_1\sqrt{r_1 + r_2}\log(1/\alpha))$ *and* $\|\mathbf{E}_t\|_F^2 \lesssim \delta^2 M_1^2(r_1 + r_2)^{1.5}\log(1/\alpha)^2.$

*Proof of Theorem 5.* The error $g$ in Lemma 15 is no less than $\Omega(\delta^2 M_1^2(r_1 + r_2)) \gg \alpha$ order, therefore $T_2 = T_1 + \frac{1}{\eta}\log(1/\alpha)$ suffices for $\sigma_{r_1}^{(t)}$ to reach $1 - 5g$. Then similar to the induction in the proof of Theorem 4, we can derive(in the same high probability event):

- $\|\mathbf{E}_t\| \leq 40\eta\delta M_1\sqrt{r_1 + r_2} + 40\eta\delta M_1\sqrt{r_1 + r_2}(t - T_1) \leq 80\delta M_1\sqrt{r_1 + r_2}\log(1/\alpha)) < 0.01;$
- $\|\mathbf{E}_t\|_F^2 \lesssim t\eta\delta^2 M_1^2(r_1 + r_2)^{1.5}\log(1/\alpha) \lesssim \delta^2 M_1^2(r_1 + r_2)^{1.5}\log(1/\alpha)^2 < 1;$
- $\|\mathbf{Q}_t\| \leq p^{-1.5}r_2^{1.5}\sqrt{r_2}\mathrm{L}_t \leq p^{-1.5}r_2^{1.5}\sqrt{r_2}40M_1\delta\sqrt{r_1 + r_2} = p^{-1.5}40M_1\delta^\star < 0.01;$

Hence the assumption in Lemma 15 is satisfied, with

$$g \lesssim p^{-3}\delta^{\star 2}M_1^2 \vee \delta M_1\sqrt{r_1 + r_2} \tag{95}$$

Therefore,

$$\left|\|\mathbf{R}_{T_2}\| - 1\|\right| \lesssim p^{-3}\delta^{\star 2}M_1^2 \vee \delta M_1\sqrt{r_1 + r_2} \tag{96}$$

$\qquad\square$

*Proof of Theorem 2.* Using Theorem 4 and Theorem 5 with $T = T_2$, we have

$$
\begin{aligned}
\|\mathbf{U}_{T_2}\mathbf{U}_{T_2}^\top - \mathbf{A}^\star\|_F &\overset{(a)}{\leq} \|\mathbf{E}_{T_2}\mathbf{E}_{T_2}^\top\|_F + \|\mathbf{R}_{T_2}^\top\mathbf{R}_{T_2} - \mathbf{I}\|_F + \|\mathbf{Q}_{T_2}^\top\mathbf{Q}_{T_2}\|_F \\
&\quad + 2\|\mathbf{E}_{T_2}\mathbf{Q}_{T_2}\|_F + 2\|\mathbf{E}_t\mathbf{R}_t\|_F + 2\|\mathbf{R}_{T_2}^\top\mathbf{Q}_{T_2}\|_F \\
&\overset{(b)}{\leq} \|\mathbf{E}_{T_2}\|_F^2 + O\left(\delta^{\star 2}M_1^2 \vee \delta M_1\sqrt{r_1+r_2}\right)\sqrt{r_1} + \|\mathbf{Q}_{T_2}\|_F^2 \\
&\quad + 2\|\mathbf{E}_{T_2}\|\|\mathbf{Q}_{T_2}\|_F + 2\|\mathbf{E}_{T_2}\|\|\mathbf{R}_t\|_F + 2\|\mathbf{Q}_{T_2}\|_F\|\mathbf{R}_t\| \\
&\overset{(c)}{\lesssim} \delta^2 M_1^2(r_1+r_2)^{1.5}\log^2(1/\alpha) + (\delta^{\star 2}M_1^2 \vee \delta M_1\sqrt{r_1+r_2})\sqrt{r_1} + \delta^{\star 2}M_1^2 \\
&\quad + (\delta^\star M_1 + (1+o(1))\sqrt{r_1})\,\delta M_1\sqrt{r_1+r_2}\log(1/\alpha) + \delta^\star M_1(1+o(1)) \\
&\lesssim (\delta^{\star 2}M_1^2\sqrt{r_1} \vee \delta^\star M_1)\log^2 d.
\end{aligned}
$$

where in $(a)$ we decompose $\mathbf{U}\mathbf{U}_t^\top$ (See (57)) and triangle inequality. In $(b)$ and $(c)$ we use Theorem 5 and repeatedly use the fact $\|\mathbf{AB}\|_F \leq \|\mathbf{A}\|\|\mathbf{B}\|_F$. This completes the proof. $\square$

# B  Deferred Proofs

## B.1  Proof of Proposition 1

In the below contexts, notations such as $C, c, C_1, c_1$ always denote some positive absolute constants. Such notation is widely adopted in the field of non-asymptotic theory.

We first state some useful definitions and lemmas:

**Definition 6** ($\epsilon$-Net and Covering Numbers)**.** *Let $(T, d)$ be a metric space. Let $\epsilon > 0$. For a subset $K \subset T$, a subset $\mathcal{M} \subseteq K$ is called an $\epsilon$-net of $K$ if every point in $K$ is within distance $\epsilon$ of some point in $\mathcal{M}$. We define the covering number of $K$ to be the smallest possible cardinality of such $\mathcal{M}$, denoted as $\mathcal{N}(K, \epsilon)$.*

**Lemma 16** (Covering Number of the Euclidean Ball)**.** *Let $\mathcal{S}^{n-1}$ denote the unit Euclidean sphere in $\mathbb{R}^n$. The following result satisfies for any $\epsilon > 0$:*

$$
\mathcal{N}(\mathcal{S}^{n-1}, \epsilon) \leq \left(\frac{2}{\epsilon} + 1\right)^n. \tag{97}
$$

**Lemma 17** (Two-sided Bound on Gaussian Matrices)**.** *Let $\mathbf{A}$ be an $d \times r$ matrix whose elements $\mathbf{A}_{ij}$ are independent $N(0,1)$ random variables. Then for any $t \geq 0$ we have*

$$
\sqrt{d} - C(\sqrt{r} + t) \leq \sigma_r(\mathbf{A}) \leq \sigma_1(\mathbf{A}) \leq \sqrt{d} + C(\sqrt{r} + t) \tag{98}
$$

*with probability at least $1 - 2\exp(-t^2)$.*

**Lemma 18** (Approximating Operator Norm Using $\epsilon$-nets)**.** *Let $\mathbf{A}$ be an $m \times n$ matrix and $\epsilon \in [0, 1/2)$. For any $\epsilon$-net $\mathcal{M}_1$ for the sphere $\mathcal{S}^{n-1}$ and any $\epsilon$-net $\mathcal{M}_2$ of the sphere $\mathcal{S}^{m-1}$, we have*

$$
\sup_{\mathbf{x}\in\mathcal{M}_1, \mathbf{y}\in\mathcal{M}_2} \langle \mathbf{Ax}, \mathbf{y}\rangle \leq \|\mathbf{A}\| \leq \frac{1}{1-2\epsilon}\sup_{\mathbf{x}\in\mathcal{M}_1, \mathbf{y}\in\mathcal{M}_2}\langle \mathbf{Ax}, \mathbf{y}\rangle. \tag{99}
$$

*Moreover, if $m = n$, then we have*

$$
\sup_{\mathbf{x}\in\mathcal{M}_1}\langle\mathbf{Ax},\mathbf{x}\rangle \leq \|\mathbf{A}\| \leq \frac{1}{1-2\epsilon-\epsilon^2}\sup_{x\in\mathcal{M}_1, y\in\mathcal{M}_2}|\langle\mathbf{Ax},\mathbf{y}\rangle|. \tag{100}
$$

**Lemma 19** (Concentration Inequality for Product of Gaussian Random Varables)**.** *Suppose $X$ and $Y$ are independent $N(0,1)$ random variables. Then $\langle X, Y\rangle$ is a sub-exponential random variable. Therefore for $(X_1, \ldots, X_m, Y_1, \ldots, Y_m)^\top \sim N(0, \mathbf{I}_{2m})$, the following holds for any $t \geq 0$:*

$$
\mathbb{P}\left(\frac{1}{m}\left|\sum_{i=1}^m \langle X_i, Y_i\rangle\right| > t\right) < 2\exp\left(-c\min(t^2, t)\cdot m\right). \tag{101}
$$

*Proof.* Note that

$$\langle X, Y \rangle = \frac{1}{2}\left(\frac{1}{\sqrt{2}}X + \frac{1}{\sqrt{2}}Y\right)^2 - \frac{1}{2}\left(\frac{1}{\sqrt{2}}X - \frac{1}{\sqrt{2}}Y\right)^2. \tag{102}$$

The two terms are independent and following Gamma distribution $\Gamma\left(\frac{1}{2}, 1\right)$. Since Gamma distribution random variables are sub-exponential, $\langle X, Y \rangle$ is sub-exponential too. The concentration inequality follows from Bernstein's inequality. (See Theorem 2.8.2 of Vershynin [47]). $\square$

Now we prove Proposition 1:

*Proof of Proposition 1.* First we provide a bound for $\|\mathbf{M}_1^\top \mathbf{M}_2\|$. We fix $\epsilon = 1/4$, and we can find an $\epsilon$-net $\mathcal{M}_1$ of the sphere $\mathcal{S}^{r_1-1}$ and $\epsilon$-net $\mathcal{M}_2$ of the sphere $\mathcal{S}^{r_2-1}$ with

$$|\mathcal{M}_1| \leq 9^{r_1}, \quad |\mathcal{M}_2| \leq 9^{r_2}. \tag{103}$$

For each $x \in \mathcal{M}_1$ and $y \in \mathcal{M}_2$, we have for $0 < u < 1$,

$$\mathbb{P}\left(\frac{1}{d}\mathbf{x}^\top \mathbf{M}_1^\top \mathbf{M}_2 \mathbf{y} > u\right) = \mathbb{P}\left(\frac{1}{d}\langle \mathbf{M}_1 \mathbf{x}, \mathbf{M}_2 \mathbf{y}\rangle > u\right) \tag{104}$$
$$\leq 2\exp(-cdu^2),$$

where we use the fact that $\mathbf{M}_1\mathbf{x}$ and $\mathbf{M}_2\mathbf{y}$ are independent $N(0, \mathbf{I}_d)$ random vectors and an application of Lemma 19. We let $u = \sqrt{\frac{r_1+r_2}{d}} \cdot t$ for $t < \sqrt{\frac{d}{r_1+r_2}}$, we have:

$$\mathbb{P}\left(\frac{1}{d}\|\mathbf{M}_1^\top \mathbf{M}_2\| \geq \sqrt{\frac{r_1+r_2}{d}} \cdot t\right) \overset{(a)}{\leq} \mathbb{P}\left(\frac{1}{d}\max_{\mathbf{x}\in\mathcal{M}_1, \mathbf{y}\in\mathcal{M}_2} \mathbf{x}^\top \mathbf{M}_1^\top \mathbf{M}_2 \mathbf{y} \geq \frac{1}{2}\sqrt{\frac{r_1+r_2}{d}} \cdot t\right)$$
$$\overset{(b)}{\leq} 9^{r_1+r_2} \cdot 2\exp\left(-c_2(r_1+r_2)t^2\right)$$
$$= 2\exp\left(-(r_1+r_2)(c_2 t^2 - \log(9))\right), \tag{105}$$

where in $(a)$ we use Lemma 18, in $(b)$ we apply a union bound over all $\mathbf{x} \in \mathcal{M}_1$ and $\mathbf{y} \in \mathcal{M}_2$.

Next, we bound $\|\mathbf{R}_1^{-1}\|$ and $\|\mathbf{R}_2^{-1}\|$. Recall the QR-decompositions of $\mathbf{M}_1$ and $\mathbf{M}_2$:

$$\mathbf{M}_1 = \mathbf{U}^\star_1 \mathbf{R}_1 \quad \text{and} \quad \mathbf{M}_2 = \mathbf{U}^\star_2 \mathbf{R}_2, \tag{106}$$

which implies $\mathbf{M}_1^\top \mathbf{M}_1 = \mathbf{R}_1^\top \mathbf{R}_1$ and $\mathbf{M}_2^\top \mathbf{M}_2 = \mathbf{R}_2^\top \mathbf{R}_2$, and consequently $\|\mathbf{R}_1^{-1}\| = \sigma_{r_1}(\mathbf{M}_1)^{-1}$ and $\|\mathbf{R}_2^{-1}\| = \sigma_{r_2}(\mathbf{M}_2)^{-1}$. From Lemma 17,

$$\mathbb{P}\left(\|\mathbf{R}_1^{-1}\| \geq \frac{2}{\sqrt{d}}\right) = \mathbb{P}\left(\sigma_{r_1}(\mathbf{M}_1) \leq \frac{\sqrt{d}}{2}\right) < 2\exp(-c_1 d). \tag{107}$$

And similarly for $\|\mathbf{R}_2^{-1}\|$. Finally, for $t < \sqrt{\frac{d}{r_1+r_2}}$,

$$\mathbb{P}\left(\left\|\mathbf{U}^{\star\top}_1 \mathbf{U}^\star_2\right\| \geq 4t\sqrt{\frac{r_1+r_2}{d}}\right)$$
$$= \mathbb{P}\left(\left\|\mathbf{R}_1^{-\top}\mathbf{M}_1^\top \mathbf{M}_2 \mathbf{R}_1^{-1}\right\| \geq 4t\sqrt{\frac{r_1+r_2}{d}}\right) \tag{108}$$
$$\leq \mathbb{P}\left(\|\mathbf{R}_1^{-1}\| \geq \frac{2}{\sqrt{d}}\right) + \mathbb{P}\left(\|\mathbf{R}_2^{-1}\| \geq \frac{2}{\sqrt{d}}\right) + \mathbb{P}\left(\frac{1}{d}\|\mathbf{M}_1^\top \mathbf{M}_2\| \geq t\sqrt{\frac{r_1+r_2}{d}}\right)$$
$$\leq 4\exp\left(-c_1 d\right) + 2\exp\left(-c_2(r_1+r_2)t^2\right).$$

This completes the proof. $\square$

## B.2 The Failure of Pooled Stochastic Gradient Descent

From Theorems 2 and 3, for the hard case in Theorem 3, we have a separation that Pooled Gradient Descent fails to select out the invariant signal, whereas the `HeteroSGD` can succeed. This isolates the implicit bias of online algorithms over heterogeneous data towards invariance and causality.

In this section, we give a rigorous proof for Theorem 3. We first demonstrate the failure of `PooledGD`

**Theorem 6** (Negative Result for Pooled Gradient Descent). *Under the assumptions of Theorem 2, for the certain case where $\mathbf{U}^\star \perp \mathbf{V}^\star$ and $\mathbb{E}_{e \in D}\mathbf{\Sigma}^{(e)} = \mathbf{I}_{r_2}$, if we perform GD over all samples from all environments and ends with $T = \Theta(\log d)$, then $\mathbf{U}_t$ keeps approaching $\mathbf{U}^\star\mathbf{U}^{\star\top} + \mathbf{V}^\star\mathbf{V}^{\star\top}$, in the sense that*

$$\left\| \mathbf{U}_T\mathbf{U}_T^\top - \mathbf{U}^\star\mathbf{U}^{\star\top} - \mathbf{V}^\star\mathbf{V}^{\star\top} \right\|_F \leq \tilde{O}(\delta^{\star^2}M_1^2\sqrt{r_1} + \delta^\star M_1) = o(1), \tag{109}$$

*during which for all $t = 0, 1, \ldots, T$:*

$$\left\| \mathbf{U}_t\mathbf{U}_t^\top - \mathbf{A}^\star \right\|_F \gtrsim \sqrt{r_1 \wedge r_2}. \tag{110}$$

*Proof of Theorem 6.* Firstly, we emphasis that Theorem 2 also applies to the case where there is only one environment and no spurious signals, the $m$ samples are generated as: (We use underlined notations to distinguish this setting from others)

$$\underline{y}_i = \langle \underline{\mathbf{X}}_i, \underline{\mathbf{A}}^\star \rangle, i = 1, \ldots, m \tag{111}$$

In such cases, there is no randomness and $\underline{\mathbf{U}}_T\underline{\mathbf{U}}_T^\top$ deterministically learns $\underline{\mathbf{A}}^\star$ and all the singular values of $\underline{\mathbf{R}}_t$ grow at similar speeds.

Under the conditions in Theorem 6, we first construct a single-environment case. Let $\mathbf{U}^\star$ and $\mathbf{V}^\star$ be defined as in Theorem 6, we let the invariant signal $\underline{\mathbf{A}}^\star = \mathbf{U}^\star\mathbf{U}^{\star\top} + \mathbf{V}^\star\mathbf{V}^{\star\top}$ and there is no spurious signal. Then the updating rule is:

$$\begin{aligned}
\underline{\mathbf{U}}_{t+1} &= \underline{\mathbf{U}}_t - \eta \left[ \frac{1}{m}\sum_{i=1}^m \langle \underline{\mathbf{X}}_i, \underline{\mathbf{U}}_t\underline{\mathbf{U}}_t^\top - \underline{\mathbf{A}}^\star \rangle \underline{\mathbf{X}}_i \right] \underline{\mathbf{U}}_t \\
&= \underline{\mathbf{U}}_t - \eta \left( \underline{\mathbf{U}}_t\underline{\mathbf{U}}_t^\top - \underline{\mathbf{A}}^\star \right) \underline{\mathbf{U}}_t - \eta \mathsf{E} \circ \left( \underline{\mathbf{U}}_t\underline{\mathbf{U}}_t^\top - \underline{\mathbf{A}}^\star \right)\underline{\mathbf{U}}_t.
\end{aligned} \tag{112}$$

Using Theorem 2, we can prove that $\underline{\mathbf{U}}_t\underline{\mathbf{U}}_t$ continuously approaches $\underline{\mathbf{A}}^{\star\top} = \mathbf{U}^\star\mathbf{U}^{\star\top} + \mathbf{V}^\star\mathbf{V}^{\star\top}$ in phase 1 & 2, during which:

- In phase 1, $\|\underline{\mathbf{R}}_t\| < 1/2$ therefore $\|\underline{\mathbf{U}}_t\underline{\mathbf{U}}_t^\top - \mathbf{U}^\star\mathbf{U}^{\star\top}\|_F \gtrsim \sqrt{r_1}$.

- In phase 2, all the singular values of $\|\underline{\mathbf{R}}_t\|$ get larger than $1/6$, from Weyl's inequality, we have that the top $r_2$ singular values of $\underline{\mathbf{U}}_t\underline{\mathbf{U}}_t^\top - \mathbf{U}^\star\mathbf{U}^{\star\top}$ are all larger than $1/6$. Hence $\|\underline{\mathbf{U}}_t\underline{\mathbf{U}}_t^\top - \mathbf{U}^\star\mathbf{U}^{\star\top}\|_F \gtrsim \sqrt{r_2}$.

Therefore, $\|\underline{\mathbf{U}}_t\underline{\mathbf{U}}_t^\top - \mathbf{U}^\star\mathbf{U}^{\star\top}\|_F \gtrsim \sqrt{r_1 \wedge r_2}$ for all $t = 0, \ldots, T$.

Now we prove Theorem 6. The updating rule can be written as

$$\begin{aligned}
\mathbf{U}_{t+1} &= \mathbf{U}_t - \eta\mathbb{E}_{e\sim D}\left[ \frac{1}{m}\sum_{i=1}^m \langle \mathbf{X}_i^{(e)}, \mathbf{U}_t\mathbf{U}_t^\top - \mathbf{A}^\star - \mathbf{A}^{(e)} \rangle \mathbf{X}_i^{(e)} \right] \mathbf{U}_t \\
&= \mathbf{U}_t - \eta \left( \mathbf{U}_t\mathbf{U}_t^\top - \mathbf{A}^\star - \mathbb{E}_{e\sim D}\mathbf{A}^{(e)} \right) \mathbf{U}_t - \eta\mathbb{E}_{e\sim D}\left[ \mathsf{E}_e \circ \left( \mathbf{U}_t\mathbf{U}_t^\top - \mathbf{A}^\star - \mathbf{A}^{(e)} \right) \right] \mathbf{U}_t \\
&= \mathbf{U}_t - \eta \left( \mathbf{U}_t\mathbf{U}_t^\top - \left( \mathbf{U}^\star\mathbf{U}^{\star\top} + \mathbf{V}^\star\mathbf{V}^{\star\top} \right) \right) \mathbf{U}_t - \eta\mathbb{E}_{e\sim D}\left[ \mathsf{E}_e \circ \left( \mathbf{U}_t\mathbf{U}_t^\top - \mathbf{A}^\star - \mathbf{A}^{(e)} \right) \right] \mathbf{U}_t.
\end{aligned}$$

We compare this updating rule with (112). The only difference is the RIP error term. However, the upper bounds for $\mathsf{E}_e \circ \left( \mathbf{U}_t\mathbf{U}_t^\top - \mathbf{A}^\star - \mathbf{A}^{(e)} \right)$ used in the proof also apply for the expectation $\mathbb{E}_{e\sim D}\left[ \mathsf{E}_e \circ \left( \mathbf{U}_t\mathbf{U}_t^\top - \mathbf{A}^\star - \mathbf{A}^{(e)} \right) \right]$. So we can derive the same conclusion that

$$\left\| \mathbf{U}_T\mathbf{U}_T^\top - \mathbf{A}^\star - \mathbb{E}_{e\sim D}\mathbf{A}^{(e)} \right\|_F \leq o(1) \tag{113}$$

for $T = \Theta(\frac{1}{\theta}\log(1/\alpha))$, during which we have that for all $t = 0, 1, \ldots, T$:

$$\left\|\mathbf{U}_t\mathbf{U}_t^\top - \mathbf{A}^\star\right\|_F \gtrsim \sqrt{r_1 \wedge r_2}. \tag{114}$$

$\square$

Now we are ready to prove Theorem 3. Assume that at each time $t = 0, \ldots, 1$, we receive $m$ samples $\{\mathbf{X}_i^{(t)}, y_i^{(t)}\}_{i=1}^m$, each sample is independently sampled from environment $e_{t,i} \sim D$, satisfying

$$y_i^{(t)} = \langle \mathbf{X}_i^{(t)}, \mathbf{A}^\star + \mathbf{A}^{(e_{t,i})}\rangle, \tag{115}$$

and imply the Stochastic Gradient Descent

$$\mathbf{U}_{t+1} = \left(\mathbf{I}_d - \eta\frac{1}{m}\sum_{i=1}^m (\langle \mathbf{X}_i^{(t)}, \mathbf{U}_t\mathbf{U}_t^\top\rangle - y_i^{(t)})\mathbf{X}_i^{(t)}\right)\mathbf{U}_t. \tag{116}$$

For technical convenience, we assume that $\mathbf{X}$ is the symmetric Gaussian matrix with diagonal elements from $N(0,1)$ and off-diagonal elements from $N(0,1/2)$. We further assume $\mathbf{X}_i^{(t)}$ is independent of $e_{t,i}$. This corresponds to the cases where each environment has infinitely many samples and the linear measurements from different environments share the same distribution.

*Proof of Theorem 3.* Denote $\bar{\mathbf{A}} = \mathbb{E}_{e\in D}\mathbf{A}^{(e)}$. Then we have

$$\mathbf{U}_{t+1} = \left(\mathbf{I}_d - \eta\frac{1}{m}\sum_{i=1}^m (\langle \mathbf{X}_i^{(t)}, \mathbf{U}_t\mathbf{U}_t^\top - \mathbf{A}^\star - \bar{\mathbf{A}}\rangle)\mathbf{X}_i^{(t)}\right)\mathbf{U}_t$$
$$+ \eta\left(\frac{1}{m}\sum_{i=1}^m \langle \mathbf{X}_i^{(t)}, \mathbf{A}^{(e_{t,i})} - \bar{\mathbf{A}}\rangle\mathbf{X}_i^{(t)}\right)\mathbf{U}_t.$$

The first term is the dynamic of single environment matrix sensing problem, and the second term is a zero-mean noise arising from SGD. Once we can prove that the second term is small with high probability, then the dynamic will be similar to the dynamic of single environment matrix sensing problem, thereby we can get a high-probability version of the result of Theorem 3.

Now we control the SGD noise term. Let $\mathcal{M}_3$ be a $\frac{1}{4}$-net of the sphere $\mathcal{S}^{d-1}$ with $|\mathcal{M}_3| \le 9^d$. Then for any $d \times d$ matrix $\mathbf{M}$, we have $\|\mathbf{M}\| \le 4\max_{\mathbf{x}\in\mathcal{M}_3} |\mathbf{x}^\top\mathbf{M}\mathbf{x}|$. For any fixed $\mathbf{x} \in \mathcal{M}_3$, one can see that $\langle\mathbf{X}_i^{(t)}, \mathbf{A}^{(e_{t,i})} - \bar{\mathbf{A}}\rangle\mathbf{x}^\top\mathbf{M}\mathbf{x}$ has zero mean and is the product of two sub-Gaussian random variable with sub-Gaussian parameter no more than $2M_1(r_1 + r_2)$ and $2$. Therefore, it is a sub-exponential random variable with parameter no more than $CM_1(r_1 + r_2)$ for some universal constant $C > 1$. Then applying the Bernstein's Inequality [47] and taking the union bound over $\mathcal{M}_3$, we can obtain that

$$\mathbb{P}\left(\sup_{\mathbf{x}\in\mathcal{M}_3} \left|\langle\mathbf{X}_i^{(t)}, \mathbf{A}^{(e_{t,i})} - \bar{\mathbf{A}}\rangle\mathbf{x}^\top\mathbf{M}\mathbf{x}\right| > CM_1(r_1 + r_2)(\sqrt{\frac{t}{m}} + \frac{t}{m})\right) < 2\cdot 9^d\exp(-t). \tag{117}$$

Setting $t = 10d$ and $m = d\operatorname{poly}(r_1 + r_2, M_1 + M_2, \log d)$, we can obtain that with probability over $1 - \exp(-d)$,

$$\left\|\frac{1}{m}\sum_{i=1}^m \langle\mathbf{X}_i^{(t)}, \mathbf{A}^{(e_{t,i})} - \bar{\mathbf{A}}\rangle\mathbf{X}_i^{(t)}\right\| \le \frac{1}{\operatorname{poly}(r_1 + r_2, M_1 + M_2, \log d)}. \tag{118}$$

Therefore, in this case the SGD error can be upper bounded in the same way as the RIP error at the level of $o(1/\operatorname{poly}(r_1 + r_2, M_1 + M_2, \log d))$. This implies that the SGD error will not significantly affect the dynamic with probability over $1 - T\exp(-d)$. Therefore (113) and (114) hold with probability over $0.99$.

$\square$

Theorem 3 and Theorem 6 indicate that the failure is because the signal is averaged when calculating gradients when we perform GD or SGD over pooled datasets. To the best of our knowledge, it is

intrinsically hard to provide a rigorous statement when the batch size is small. We would like to leave the theoretical analysis as a future work. In the following simulation, we aim to demonstrate empirically that Pooled SGD fails to learn invariance with a small batch size. We consider the $|\mathcal{E}| = 2$ case and the environments are generated by $\mathbf{A}^{(1)} = \mathbf{U}^\star \mathbf{U}^{\star\top} + (s + M)\mathbf{V}^\star \mathbf{V}^{\star\top}$ and $\mathbf{A}^{(2)} = \mathbf{U}^\star \mathbf{U}^{\star\top} + (s - M)\mathbf{V}^\star \mathbf{V}^{\star\top}$ where $(\mathbf{U}^\star, \mathbf{V}^\star)$ is column orthonormal. Then the invariant solution is $\mathbf{A}^\star = \mathbf{U}^\star \mathbf{U}^{\star\top}$ and the spurious solution is $\mathbf{A}^\star + \bar{\mathbf{A}} = \mathbf{U}^\star \mathbf{U}^{\star\top} + s\mathbf{V}^\star \mathbf{V}^{\star\top}$. We set $(\alpha, d, r_1, r_2, s, M, m) = (10^{-3}, 30, 5, 5, 0.5, 4, 80)$, use Gaussian measurements as Section 5 and let $T$ be sufficiently large. The following shows the F-norm between $\mathbf{U}_t \mathbf{U}_t^\top$ and $\mathbf{A}^\star$ or $\mathbf{A}^\star + \bar{\mathbf{A}}$.

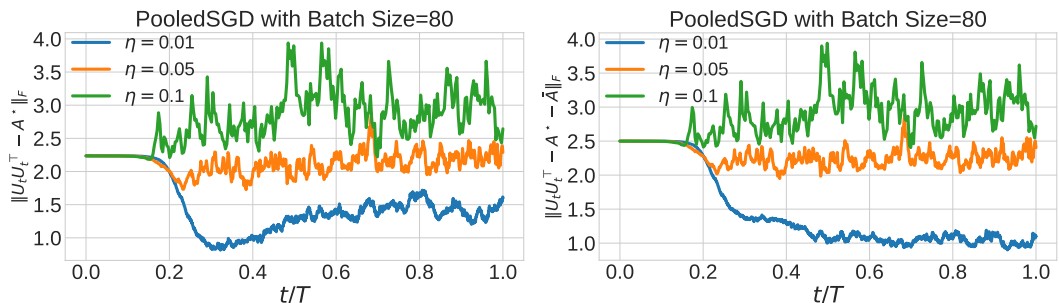

Figure 6: These figures shows that when the batch size is small, the trajectory will be far away from $\mathbf{A}^\star$ and $\mathbf{A}^\star + \bar{\mathbf{A}}$, suggesting that the algorithm is not stable in this regime.

## C   Neural Networks with Quadratic Activations

In this section we discuss how to apply our results to nerral networks with quadratic activations. In particular, Example 1. As discussed above,

$$y_i^{(e)} = \sum_{j=1}^{r_1} q(\mathbf{a}_j^\top \mathbf{x}_i^{(e)}) + \sum_{j=r_1+1}^{r} a_j^{(e)} q(\mathbf{a}_j^\top \mathbf{x}_i^{(e)}) = \left\langle \mathbf{x}_i^{(e)} \mathbf{x}_i^{(e)\top}, \sum_{j=1}^{r_1} \mathbf{a}_j \mathbf{a}_j^\top + \sum_{j=r_1+1}^{r} a_j^{(e)} \mathbf{a}_j \mathbf{a}_j^\top \right\rangle,$$
(119)

and it is equivalent to matrix sensing problem with

$$\mathbf{A}^\star = \sum_{j=1}^{r_1} \mathbf{a}_j \mathbf{a}_j^\top \,, \; \mathbf{A}^{(e)} = \sum_{j=r_1+1}^{r} a_j^{(e)} \mathbf{a}_j \mathbf{a}_j^\top \text{ and } \mathbf{X}_i^{(e)} = \mathbf{x}_i^{(e)} \mathbf{x}_i^{(e)\top}.$$
(120)

The main difference is that, when the samples $\mathbf{x}_i$ are i.i.d. $N(0, \mathbf{I}_d)$, the set of linear measurements $\{\mathbf{x}_1 \mathbf{x}_1^\top, \dots, \mathbf{x}_m \mathbf{x}_m^\top\}$ no longer satisfies the RIP property. However, the following lemma tells that, with proper truncation, the set of measurements enjoys similar properties.

**Lemma 20** (Lemma 5.1 of Li et al. [31]). *Let* $(\mathbf{X}_1, \dots, \mathbf{X}_m) = \{\mathbf{x}_1 \mathbf{x}_1^\top, \dots, \mathbf{x}_m \mathbf{x}_m^\top\}$ *where* $\mathbf{x}_i$*'s are i.i.d.* $\sim \mathcal{N}(0, \mathbf{I})$*. Let* $R = \log\left(\frac{1}{\delta}\right)$*. Then, for every* $q, \delta \in [0, 0.01]$ *and* $m \gtrsim d \log^4 \frac{d}{q\delta}/\delta^2$*, with probability at least* $1 - q$*, we have that for every symmetric matrix* $\mathbf{A}$*:*

$$\left\| \frac{1}{m} \sum_{i=1}^{m} \langle \mathbf{X}_i, \mathbf{A} \rangle \mathbf{X}_i 1_{|\langle \mathbf{X}_i, \mathbf{A} \rangle| \leq R} - 2\mathbf{A} - \mathrm{tr}(\mathbf{A})\mathbf{I} \right\| \leq \delta \|\mathbf{A}\|_\star.$$
(121)

*If* $\mathbf{A}$ *has rank at most* $r$ *and operator norm at most* 1*, we have:*

$$\left\| \frac{1}{m} \sum_{i=1}^{m} \langle \mathbf{X}_i, \mathbf{A} \rangle \mathbf{X}_i 1_{|\langle \mathbf{X}_i, \mathbf{A} \rangle| \leq R} - 2\mathbf{A} - \mathrm{tr}(\mathbf{A})\mathbf{I} \right\| \leq r\delta.$$
(122)

To accommodate this difference, we adopt the modified version of loss function and algorithm from Li et al. [31].

**Algorithm 4** Modified Algorithm For Neural Network with Quadratic Activations

---

Set $\mathbf{U}_0 = \alpha \mathbf{I}_d$, where $\alpha$ is a small positive constant.
Set step size $\eta$.
**for** $t = 1, \ldots, T - 1$ **do**
    Receive $m$ samples $(\mathbf{x}_i^{(e_t)}, y_i^{(e_t)})$ from current environment $e_t$.
    Calculate $\hat{y}_i^{(e_t)} = \mathbf{1}^\top q(\mathbf{U}_t \mathbf{x}_i^{(e_t)})$, $i = 1, 2, \ldots, m$.
    Calculate modified loss function $\tilde{\mathcal{L}}_t(\mathbf{U}_t) = \frac{1}{m} \sum_{i=1}^m \left( \hat{y}_i^{(e_t)} - y_i^{(e_t)} \right)^2 \mathbf{1}_{\|\mathbf{U}^\top \mathbf{x}_i^{(e_t)}\|^2 \leq R}$
    Gradient Descent $\tilde{\mathbf{U}}_t = \mathbf{U}_t - \eta \nabla \tilde{f}_\mathcal{L}(\mathbf{U}_t)$.
    Let $\tau_t = \|\mathbf{A}^\star + \mathbf{A}^{(e_t)}\|$.
    Shrinkage $\mathbf{U}_{t+1} = \frac{1}{1 - \eta(\|\mathbf{U}_t\|_F^2 - \tau_t)} \tilde{\mathbf{U}}_t$
**end for**
**Output:** $\mathbf{U}_T$.

---

**Remark 1.** *Here we encounter the same caveat that Algorithm 4 requires our knowledge on $\tau_t$. As discussed in Li et al. [31], the algorithm is likely to be robust if $\tau_t$ is replaced by its moment estimation.*

Now we outline the proof sketch of Example 1

**Theorem 7** (Two-Layer NN with Quadratic Activation). *Let $\mathbf{a}_1, \cdots, \mathbf{a}_r \in \mathbb{R}^d$ be independent random vectors sampled from normal distribution $N(0, \frac{1}{d}\mathbf{I}_d)$. For environment $e \in \mathcal{E}$, suppose the target function is determined by $r_1$ invariant features and $r_2$ variant admits that for each sample $(\mathbf{x}_i^{(e)}, y_i^{(e)})$:*

$$y_i^{(e)} = \sum_{j=1}^{r_1} q(\mathbf{a}_j^\top \mathbf{x}_i^{(e)}) + \sum_{j=r_1+1}^{r} a_j^{(e)} q(\mathbf{a}_j^\top \mathbf{x}_i^{(e)}) = \left\langle \mathbf{x}_i^{(e)} \mathbf{x}_i^{(e)\top}, \sum_{j=1}^{r_1} \mathbf{a}_j \mathbf{a}_j^\top + \sum_{j=r_1+1}^{r} a_j^{(e)} \mathbf{a}_j \mathbf{a}_j^\top \right\rangle. \quad (123)$$

*Suppose we train the following two-layer NN:*

$$f(\mathbf{x}) = \sum_{j=1}^d q(\mathbf{u}_j \mathbf{x}), \quad (124)$$

*and the initialization of parameters $\{\mathbf{u}_j\}$ satisfies $\sum_{j=1}^d \mathbf{u}_j \mathbf{u}_j^\top = \alpha \mathbf{I}$. If $\{a_j^{(e)}\}_{j,e}$ satisfies $\frac{\sup_{e,j}\{|a_j^{(e)}|\} \cdot \max_j\{1 + |\mathbb{E}_e a_j^{(e)}|\}}{\min_j\{\mathrm{Var}_e[a_j^{(e)}]\}} < c_0$ for some absolute constant $c_0$, sample complexity $m \gg d \,\mathrm{poly}(r, \log(d), \sup_{e,j}\{|a_j^{(e)}|\})$, $\alpha \in (d^{-4}, d^{-1})$ and $\eta \sim \frac{\max_j\{1 + |\mathbb{E}_e a_j^{(e)}|\}}{\min_j\{\mathrm{Var}_e[a_j^{(e)}]\}}$, then Algorithm 4 returns solution that satisfies*

$$\|\sum_{j=1}^d \mathbf{u}_j \mathbf{u}_j^\top - \mathbf{A}^\star\|_F < o(1) \quad (125)$$

*with probability over 0.99.*

*Proof.* similar to the proof of Theorem 1.2 of Li et al. [31], the modified algorithm is in fact equivalent to (21) with RIP parameter $(r, \delta)$ when $m = \tilde{\Omega}(dr^2\delta^{-2})$. Hence it is fully reduced to the matrix sensing problem.

Now we verify the conditions for $\mathbf{A}^\star = \sum_{j=1}^{r_1} \mathbf{a}_j \mathbf{a}_j^\top$ and $\mathbf{A}^{(e)} = \sum_{j=r_1+1}^{r} a_j^{(e)} \mathbf{a}_j \mathbf{a}_j^\top$. Since $\mathbf{u}_i, i = 1, \ldots, r$ are independently and uniformly sampled from sphere, we have that

- With high probability over the randomness of $\{\mathbf{a}_i\}_i$, the eigenvalues of $\mathbf{A}^\star$ lie within $\left(1 - O(\sqrt{r_1}/\sqrt{d}), 1 + O(\sqrt{r_1}/\sqrt{d})\right)$ (See Theorem 4.6.1 of Vershynin [47]).

- The angle between $\mathrm{Col}(\mathbf{A}^\star)$ and $\mathrm{Col}(\mathbf{A}^{(e)})$ is of $O(\sqrt{r_1 + r_2}/\sqrt{d})$ order.

Therefore we can construct two column orthogonal matrix $\mathbf{U}^\star$ and $\mathbf{V}^\star$ such that $\mathbf{U}^{\star\top}\mathbf{V}^\star = 0$ and $\sin(\mathrm{col}(\mathbf{U}^\star), \mathrm{col}(\mathbf{A}^\star)), \sin(\mathrm{col}(\mathbf{V}^\star), \mathrm{col}(\mathbf{A}^{(e)})) \lesssim \sqrt{r_1 + r_2}/\sqrt{d}$. Hence we can apply Theorem 2 on $\tilde{\mathbf{A}}^\star := \mathbf{U}^\star\mathbf{U}^{\star\top}$ and $\tilde{\mathbf{A}}^{(e)} := \mathbf{V}^\star\mathrm{diag}(a_i^{(e)})\mathbf{V}^{\star\top}$. Such approximation only raises $O(\sqrt{r_1 + r_2}/\sqrt{d})$ multiplicative error, which is negligible. And we can easily verify $\tilde{\mathbf{A}}^{(e)}$ satisfies Assumption 2. Then this result follows from the proof of Theorem 9. $\qquad\square$

# D  The $\kappa(\mathbf{A}^\star) > 1$ Case

In this section we show how to generalize our results to the $\kappa(\mathbf{A}^\star) > 1$ case by leveraging the adaptive subspace technique proposed by Li et al. [31] for single environment setting. This framework mainly consists of the following steps:

First, instead of using the fixed subspace $\mathrm{col}(\mathbf{U}^\star)$, we use an adaptive one $S_t$, where $S_0 = \mathrm{col}(\mathbf{U}^\star)$ and $S_{t+1} = (\mathbf{I} - \eta\mathbf{M}_t)S_t$ where $\mathbf{M}_t = \frac{1}{m}\sum_{i=1}^m \langle \mathbf{X}_i, \mathbf{U}_t\mathbf{U}_t^\top - \mathbf{A}^\star\rangle\mathbf{X}_i$. And we denote $\mathbf{Z}_t = \mathrm{Id}_{S_t}\mathbf{U}_t$ and $\mathbf{H}_t = (\mathrm{Id} - \mathrm{Id}_{S_t})\mathbf{U}_t$. Which makes the updating of $\mathbf{H}_t$ substantially disentangled from $\mathbf{Z}_t$.

Second, we reason about the updating rule of $\mathbf{Z}_t$. Since the subspace is updated at each step, the updating rule of $\mathbf{Z}_t$ becomes indirect. We introduce $\tilde{\mathbf{Z}}_t = (\mathrm{Id} - \eta\mathbf{H}_t\mathbf{Z}_t^\top)\mathbf{Z}_t(\mathrm{Id} - 2\eta\mathbf{Z}_t^+\mathrm{Id}_{S_t}\mathbf{M}_t\mathbf{H}_t)$ so that $\mathbf{Z}_{t+1} \approx \tilde{\mathbf{Z}}_t - \eta\nabla\mathcal{L}(\tilde{\mathbf{Z}}_t)$. It can be shown $\sigma_{\min}(\mathbf{Z}_t)$ continually increases until it gets larger than $\frac{1}{2\sqrt{\kappa}}$.

During this iteration, we can keep $\tilde{\mathbf{Z}}_t$ is near $\mathbf{Z}_t$ for each $t$ and the principal angle $\theta_t$ between $S_t$ and $\mathrm{col}(\mathbf{U}^\star)$ satisfies $\sin(\theta_t) \lesssim \eta\rho t$ where $\rho = \tilde{\Theta}(\frac{\delta\sqrt{r}}{\kappa})$.

Finally, when $\sigma_{\min}(\mathbf{Z}_t)$ is sufficiently large and principal angle is small, we can use the local restricted strongly convex property of $\mathcal{L}$ around $\mathbf{A}^\star$ to prove $\|\mathbf{U}_t\mathbf{U}_t\|_F^2$ converges with rate $1 - \Theta(\eta/\kappa)$.

For the multi-environment setting, we have the following result under a slightly stronger assumption on the heterogeneity:

**Theorem 8** (General Theorem). *Under Assumption 1 and 2, suppose the heterogeneity parameter $M_2 \gtrsim r^2$, $\epsilon_1 < \delta$. and the RIP parameter $\delta \lesssim \frac{1}{\mathrm{poly}(r,\log(d),M_1+M_2,\kappa)}$. We choose the $\eta \in (24M_2^{-1}, \frac{1}{64}M_1^{-1} \wedge \frac{1}{r^2})$ and $\alpha \in (1/d^4, 1/d^3)$, then running Algorithm 3 in $T = \Theta(\log(\alpha^{-1})/\eta)$ steps, the algorithm outputs $\mathbf{U}_T$ that satisfies*

$$\|\mathbf{U}_T\mathbf{U}_T^\top - \mathbf{A}^\star\|_F \leq o(1) \tag{126}$$

*with probability over 0.99.*

Since the full proof of the adaptive subspace technique is involved, for clear representation, we point out the main differences from the single-environment case. We need to address the following three issues: (1) How to introduce the spurious component $\mathbf{Q}_t$ into the original framework; (2) Whether the spurious signal $\mathbf{A}^{(e)}$ significantly perturbs the dynamic of $\mathbf{Z}_t$; and (3) How to give a phase 2 analysis when there is no local restricted strongly convexity around $\mathbf{A}^\star$?

We first cope with (1). With abuse of notation, we adopt the $\mathbf{M}_t, \mathbf{Z}_t, \mathbf{H}_t$ and additionally define $\mathbf{V}_t^{(ada)} = \mathrm{Id}_{\mathbf{V}^\star}\mathbf{H}_t$ and $\mathbf{E}_t^{(ada)} = (\mathrm{Id} - \mathrm{Id}_{\mathbf{V}^\star})\mathbf{H}_t$. We can prove that

$$
\begin{aligned}
\mathbf{V}_{t+1}^{(ada)} &= \left(\mathrm{Id}_{\mathbf{V}^\star} + \eta\mathbf{A}^{(e_t)} + O(\eta\delta\sqrt{r}M_1 + (1 + \eta M_1)\sin(\theta_t))\right)\left(\mathbf{V}_t^{(ada)} + \mathbf{E}_t^{(ada)}\right) \\
&\approx \left(\mathrm{Id}_{\mathbf{V}^\star} + \eta\mathbf{A}^{(e_t)}\right)\mathbf{V}_t^{(ada)} + \text{small terms}, \\
\mathbf{E}_{t+1}^{(ada)} &= \left(\mathrm{Id}_{\mathrm{res}} + O(\eta\delta\sqrt{r}M_1 + (1 + \eta M_1)\sin(\theta_t))\right)\left(\mathbf{V}_t^{(ada)} + \mathbf{E}_t^{(ada)}\right). \\
&\approx \mathbf{E}_t^{(ada)} + \text{small terms}.
\end{aligned} \tag{127}
$$

If we can ensure $\sin(\theta_t) \lesssim \delta\,\mathrm{poly}(r, M_1 + M_2, \log(d))$, we can get similar dynamics as (23) and 24, then apply similar techniques in Section A.4 and Section A.5 to ensure $\mathbf{V}_t$ and $\mathbf{E}_t$ are no more than $\delta\,\mathrm{poly}(r, M_1 + M_2, \log(d))$. w.h.p. Moreover, the dynamics in (127) is multiplicative, which means if we decrease $\alpha$ by comparing to Theorem 2, $\mathbf{V}_t$ and $\mathbf{E}_t$ can be further upper bounded by $d^{-1}\delta\,\mathrm{poly}(r, M_1 + M_2, \log(d))$ in phase 1.

For issue (2), the spurious signal $\mathbf{A}^{(e)}$ brings error about $1 + O((\delta + \epsilon_1)\sqrt{r}M_1+)$ multiplicative factor, which can be absorbed by the inherent RIP error of $\mathbf{A}^\star$. Another difference is that, at the beginning $\|\mathbf{V}_t\|$ or $\|\mathbf{E}_t\|$ may be substantially larger than $\mathbf{Z}_t$ due to the oscillation. We emphasis that such interference happens in RIP error term or non-orthogonal error term, multiplied by $\delta, \sin(\theta_t)$ or $\epsilon_1$. We can ensure such interference is negligible when $\delta \lesssim \frac{1}{\operatorname{poly}(r, \log(d), M_1 + M_2, \kappa)}$. Therefore, the dynamic of $\mathbf{Z}_t$ is benign, and the principal angle can be bounded by $\delta \operatorname{poly}(r, \log(d), M_1 + M_2, \kappa) \ll 1$.

Finally for issue (3), when $\sigma_{\min}(\mathbf{Z}_t) \geq \frac{1}{2\sqrt{\kappa}}$ and $\sin(\theta_t) = \delta \operatorname{poly}(r, \log(d), M_1 + M_2, \kappa) \ll 1$. We get back to the original subspace $\operatorname{col}(\mathbf{U}^\star)$ and $\operatorname{col}(\mathbf{V}^\star)$. We have $\mathbf{U}_t = \mathbf{Z}_t + O(\sin(\theta_t))$, $\mathbf{V}_t = \mathbf{V}_t^{(ada)} + O(\sin(\theta_t))$, $\mathbf{E}_t = \mathbf{E}_t^{(ada)} + O(\sin(\theta_t))$ and $\|\mathbf{E}_t - \mathbf{E}_t^{(ada)}\|_F \lesssim \sqrt{r}\sin(\theta_t)$. Then we can use the technique from phase 2 analysis (Theorem 5) to complete the proof. We leave the extension of this theorem for the case where $M_1, M_2$ are constant level for future studies.

