# OpenReview forum: "The Implicit Bias of Heterogeneity towards Invariance: A Study of Multi-Environment Matrix Sensing"
_NeurIPS.cc/2024/Conference — NeurIPS 2024 poster_

### Official Review · Reviewer_hg76 · 2024-07-12

**Soundness:** 4
**Presentation:** 4
**Contribution:** 3
**Rating:** 7
**Confidence:** 2

**Summary:**

This paper studies "implicit invariance learning" within a simplified but meaningful setting---multi-environment low-rank matrix sensing problem. Authors show the implicit bias of SGD over heterogeneous data drives the model learning towards an invariant solution. The key insight is, through simply employing the large step size large-batch SGD sequentially in each environment without any  explicit regularization, the oscillation caused by heterogeneity can provably prevent model learning spurious signals, while the model learned using pooled GD over all data would simultaneously learn both the invariant and spurious signals.

**Strengths:**

- A novel perspective studying the invariant learning problem over multi environments.

- Rigorous theoretic results with detailed proof. The theoretic findings are potentially very interesting and helpful.

- Presentation is very good, and I am enjoy reading through the submission

**Weaknesses:**

The considered setting is somewhat simplified, but that is fine for the first work from this new perspective.

**Questions:**

To clarify, I am not very familiar with implicit regularization of SGD and I do not check carefully the proof. My confidence score should be low.

---

From my sensible review (assuming correctness of proof), this submission is a clear accept, so my review would be short.

- regarding: "In this paper, we will show that surprisingly, if each batch is sampled from data in one environment rather than data in all the environments, the heterogeneity in the environments together with the implicit regularization effects in the SGD algorithm can drive ..."

  This finding is interesting and potentially quite helpful to invariant learning. From my own experience with many DG problems, I like to collect training data from each environment in a large batch, and this simple method serves a good baseline and usually has some improvement over ERM. This observation was also summarized in a paper by other researchers ("Simple data balancing achieves competitive worst-group-accuracy", CLEAR 2022), but is different from here. Of course,  the above observation is only empirical and does not necessarily hold in general. Nevertheless, I would like to see some discussions with this observation regarding more general representation-learning based DG problems. I also hope this information can help authors futher consider similar theoretic studies in the more general DG setting, and I look forward to new findings in expanded version or new works.

- I believe the setting of applying SGD to varying environment should have been considered in the context of federated learning. Can you also provide additional discussion in your related work part?

- minor suggestion:

  - I don't think it necessary to put a summary in the abstract.;
  - line 22:  ”??" appear after "conditions:".

**Limitations:**

see above.

---

> ### Author Rebuttal · Authors · 2024-08-04
>
> We would like to thank the reviewers for the valuable feedback and insightful comments. We have carefully considered your comments and questions and have addressed them as below:
>
> > This finding is interesting and potentially quite helpful to invariant learning. From my own experience with many DG problems, I like to collect training data from each environment in a large batch, and this simple method serves a good baseline and usually has some improvement over ERM. This observation was also summarized in a paper by other researchers ("Simple data balancing achieves competitive worst-group-accuracy", CLEAR 2022), but is different from here. Of course, the above observation is only empirical and does not necessarily hold in general. Nevertheless, I would like to see some discussions with this observation regarding more general representation-learning based DG problems. I also hope this information can help authors further consider similar theoretic studies in the more general DG setting, and I look forward to new findings in expanded version or new works.
>
> A: Thanks for sharing this point. These empirical experiences strongly motivate us to consider generalizing our findings to other distribution-agnostic settings or models. The mentioned paper utilizes group reweighting and subsampling. It is indeed an effective approach in invariant learning contexts.  We will add some additional discussion in the revised version.
>
> > I believe the setting of applying SGD to varying environment should have been considered in the context of federated learning. Can you also provide additional discussion in your related work part?
>
> A: Thanks for pointing out this. Indeed federated learning naturally possesses the multi-environment structure. We would like to add some discussion about the relation to the works in federated learning. Federated learning ([1,2]) is a machine learning paradigm where data is stored separately and locally on multiple clients and not exchanged, and clients collaboratively train a model. Extensive work has focused on designing effective decentralized algorithms (e.g. [1,3]) while preserving privacy (e.g. [4,5]). The importance of fairness in federated learning has also garnered attention ([6,7]). One important issue in federated learning is to handle the heterogeneity across the data and hardware. Our work shows that by training with certain stochastic gradient descent methods, the system can automatically remove the bias from the individual environment and thus learn the invariant features. Our work provides insights into discovering the implicit regularization effects of standard decentralized algorithms. More discussion and related work will be added in the revised version.
>
> [1] Communication-efficient learning of deep networks from decentralized data.
>
> [2] Advances and Open Problems in Federated Learning
>
> [3] Scaffold: Stochastic controlled averaging for federated learning.
>
> [4] Our data, ourselves: Privacy via distributed noise generation.
>
> [5] On the upload versus download cost for secure and private matrix multiplication
>
> [6] Ditto: Fair and robust federated learning through personalization.
>
> [7] Personalized Federated Learning towards Communication Efficiency, Robustness and Fairness
>
> ----
>
> We again thank you for dedicating your time and effort to reviewing our manuscript. We appreciate that you highly acknowledge our work, and your suggestions will greatly help us improve our work. We will also carefully revise the paper according to your minor suggestions.

---

> > ### Comment · Reviewer_hg76 · 2024-08-08
> >
> > Thanks for your response. Look forward to expanded works on more general settings.

---

### Official Review · Reviewer_a5nG · 2024-07-12

**Soundness:** 3
**Presentation:** 3
**Contribution:** 3
**Rating:** 7
**Confidence:** 3

**Summary:**

The authors show that in a matrix sensing context, and under a data distribution that includes invariant and environment-dependent components, SGD with successive batches from different environments lead to invariant features being provably learned. The authors show that SGD with mixed batches provably does not learn the invariant solution.

**Strengths:**

- The paper addresses an important issue, the problem of learning invariant solutions, and targets a particular, well defined sub-problem within this issue.
- The assumptions that allow the authors' exposition are clearly stated and well-justified, and their results justify the claims made.
- The authors' argumentation is easy to follow and their results are situated well within the relevant literature.

**Weaknesses:**

- The authors limit their analysis to the setting of matrix sensing (and a 2-layer NN that is constructed to conform to it). This is a reasonable choice to enable their analyses, however the justifications and implications of this choice should be discussed more thoroughly.
- Although their overall argumentation is easy to follow, certain gaps and/or inconsistencies in their exposition present difficulties (see below for details).

**Questions:**

- The authors results constitute a strong negative finding w.r.t. standard SGD-based training without environment annotation and presumably mixed minibatches. The authors should more clearly highlight this and explain the reasons for the success of standard SGD-based training in practical settings (_despite_ their findings).
- Do the authors' results imply that in any dataset, even without access to environment labels, training with SGD with batch size=1 allows access to the favorable results they present in Thm 1? This is due to a single sample inevitably belonging to a single environment.
- L47: "Learning invariant predictions produces reliable, fair, robust predictions against strong structural mechanism perturbation." Please be more clear and specific.
- L53: Typo: "may not necessarily in practice"
- L58: Please describe matrix sensing problem either in the main paper, or refer to a relevant appendix section
- L61-64: The introductory exposition is more confusing then helpful since the notation is not sufficiently introduced.
- Similar to above, Figure 1 is a lavish use of authors' space without a strong unique contribution
- L140: Please fix typo (combine sentences)
- L152: Total dimension of spurious signals or total dimension of core + spurious signals?
- L199: That the spurious features have large dispersion across environments is an important assumption for the present paper. Please provide positive and negative hypothetical examples for this in realistic settings (in what cases it is reasonable to expect such large heterogeneity? In what cases it is not?)
- L200: Missing ref.
- L280: The terms HeteroSGD and PooledSGD are proposed early in the paper but later neglected, please consistently refer to the algorithms as such for readability

**Limitations:**

The limitations implied by the choice of matrix sensing setting is not sufficiently discussed.

---

> ### Author Rebuttal · Authors · 2024-08-04
>
> We would like to thank the reviewers for the valuable feedback and insightful comments. We have carefully considered your comments and questions and have addressed them as below:
>
> > About discussing the adoption of matrix sensing problem.
>
> A: We fully understand your concerns regarding our model. We adopt the matrix sensing problem because it is widely used in implicit regularization contexts, and its non-convexity and over-parameterized nature reflects the complexity of deep learning models. From this perspective, this model is sufficient for us to convey our insights. We are willing to generalizing our findings to more models in future work.
>
> > Do the authors' results imply that in any dataset, even without access to environment labels, training with SGD with batch size=1 allows access to the favorable results they present in Thm 1? This is due to a single sample inevitably belonging to a single environment.
>
> A: This is a very good point. Our results **do not imply** that the ``HeteroSGD`` succeeds when batch size=1, as our results relies on the satisfaction of the RIP condition. It is interesting to study how small batch sizes affect the invariance learning for SGD. This problem requires much more involved analysis to deal with the randomness of data when the RIP is broken down. We will discuss the main difficulty in the revised version and leave the analysis as a future work.
>
>
> > About the presentation issues.
>
> A: Thanks for pointing out these issues. We have carefully resolved the presentation issues in the revised version.
>
> ----
>
> We again thank you for dedicating your time and effort to reviewing our manuscript. Your valuable questions and insightful comments have significantly improved our work. We hope our responses have addressed all your concerns.

---

> > ### Comment · Reviewer_a5nG · 2024-08-12
> >
> > I thank the authors for their response and the additional discussion, I believe the modifications they committed to will improve the paper.

---

### Official Review · Reviewer_JQ35 · 2024-07-15

**Soundness:** 2
**Presentation:** 3
**Contribution:** 3
**Rating:** 6
**Confidence:** 3

**Summary:**

The paper studies the implicit bias of (Hetero) SGD towards learning invariant representations and discarding non-invariant features in a matrix sensing setup. The applicability of the setup to the 2-layer NN with quadratic activations is demonstrated. Finally, the authors demonstrate that pooledGD fails to recover the corresponding invariant representation. Experiments on synthetic data is used to demonstrate the validity of the claims.

**Strengths:**

- The paper is clearly written, and the motivation is clear.
- The theoretical analysis seems sound.

**Weaknesses:**

- Claims for PooledSGD: The introduction claims that the authors show that PooledSGD fails in recovering the invariant representation. However, the analysis (theorem 3) and the experiments are only presented for PooledGD. It is quite well known that GD, when compared to SGD, results in worse generalization (see [1] and references therein), even in the homogeneous environment setting. Thus, given the context, the negative result for PooledGD is not very surprising. Additionally, the assumption in theorem 3 about that the expectation of the covariance matrix, that the training environments span all directions uniformly, seems rather restrictive.
 - Theorem 2 feasibility: The application of theorem 2 in eq. 10 requires assuming d log^2(d) << m (batch size), which seems unrealistic in standard machine learning scenarios. The authors can perhaps make the point more convincingly by demonstrating the applicability of the results of theorem 2 with more realistic values (d, m, C etc.).

References:
- [1] On the Generalization Benefit of Noise in Stochastic Gradient Descent - Smith et al, 2020

**Questions:**

[1] Does the result in theorem 3 extend to pooledSGD as well?

[2] Typo: Missing Reference - line 200

**Limitations:**

Yes

---

> ### Author Rebuttal · Authors · 2024-08-04
>
> We would like to thank the reviewers for the valuable feedback and insightful comments. We have carefully considered your comments and questions and have addressed them as below:
>
> > About the findings on ``HeteroSGD`` verses ``PooledGD``? Does the result in theorem 3 extend to pooledSGD as well?
>
> A: Thanks for raising this. Our key point is that the separation comes from ``Hetero``, not from ``SGD`` itself. In fact, we can prove that ``PooledSGD`` will fail to learn the invariant signal as well as ``PooledGD``. The theorem is as follows, which will appear in the revised version of our paper:
>
> Theorem 3.1 (Negative Result for ``PooledSGD``). Under the assumptions of Theorem 3, if we perform SGD ending with $T=\Theta(\log d)$,  where each sample of a batch is randomly chosen from an environment w.r.t. $D$, and the linear measurements are symmetric Gaussian, and the batch size is $d\cdot\mathrm{poly}(r_1,r_2,M_1, \log(d))$, then with probability over 0.99,
> $$
> \left\| \mathbf{U}_T\mathbf{U}_T^\top - \mathbf{U}^*{\mathbf{U}^*}^\top-\mathbf{V}^*{\mathbf{V}^*}^\top \right\|_F = o(1),
> $$
> during which for all $t=0,1,\ldots, T$:
> $$
> \left\| \mathbf{U}_t\mathbf{U}_t^\top - \mathbf{A}^*\right\|_F\gtrsim \sqrt
> {r_1\wedge r_2}.
> $$
> The intuition is that the heterogeneity across environment is reduced. Hence the gradients will be very close to the gradients in ``pooledGD``. We remark that the counterpart for small batch size case even for single-environment case, to the best of our knowledge.
>
> > About the feasibility of Theorem 2 and assumptions.
>
> A: We fully understand your concerns regarding the assumptions. In our results, we set the batch size to satisfy the **RIP condition**, so that we can omit the randomness within the environment and focus on the randomness of the sampled environments. This condition is very common in the field of matrix sensing. However, it is interesting to study how small batch sizes affect the invariance learning for SGD. This problem requires much more involved analysis to deal with the randomness of data when the RIP is broken down. However, in practice, one does not often need such a large size of data to reach the conditions because the analysis is in the worst case. We will discuss the main difficulty in the revised version and leave the analysis as a future work.
>
> As for the assumptions, in section D we briefly show how to generalize our results to the $\kappa(\mathbf A^*)>1$ case. In the main text we adopt this for clear presentation, since this involved technique mainly works for the invariance part $ \mathbf{R}_t$, while we are more interested in the analysis of the **spurious part** $\mathbf{Q}_t$.
>
> ----
>
> We again thank you for dedicating your time and effort to reviewing our manuscript. Your valuable questions and insightful comments have significantly improved our work. We hope our responses have addressed all your concerns.

---

> ### Comment · Reviewer_JQ35 · 2024-08-12
> **Response to Rebuttal**
>
> Clarifications Needed:
> - I thank the authors for their response and clarifications. Regarding the first point, could you please provide a quick clarification for the following statement:
>
> > We remark that the counterpart for small batch size case even for single-environment case, to the best of our knowledge.
>
> - And just to confirm, for theorem 3.1 on PooledSGD, I presume the authors still require the large batch size for the RIP condition to hold/gradients to behave similar to PooledGD?
>
> Overall, I am unsure about the extent of the applicability of the matrix sensing paradigm to standard heterogeneous learning scenarios. However, I do believe the paper offers novel theoretical insights. Thus, I opt to increase my score by 1 point.
>
> As a minor suggestion, even if the theoretical analysis is too involved for this work, I would request the authors to include numerical simulations for cases when some of the assumptions (eg. batch size) are violated, in order to help the wider audience better understand any potential limitations of this perspective.

---

> > ### Author Response · Authors · 2024-08-12
> >
> > We thank the reviewer for the additional comments and are sorry for the confusing part of the response.
> >
> > > Regarding the first point, could you please provide a quick clarification for the following statement ...
> >
> > For statement, it should be
> >
> > We remark that in theory, it is still open whether PooledSGD can attain the same goal of HeteroSGD using small $m \ll d \mathrm{poly}(r_1,r_2, M_1, log(d))$, to the best of our knowledge.
> >
> > This is because under this regime, running gradient descent may not converge and thus it is hard to characterize its behavior.  We ran some simulations and found when a small batch size is adopted for PooledSGD, its gradient descent trajectory is far away from both the invariant solution $U^* (U^*)^\top$ and the pooled solution $U^* (U^*)^\top + V^* (V^*)^\top$.
> >
> > > And just to confirm, for theorem 3.1 on PooledSGD, I presume the authors still require the large batch size for the RIP condition to hold/gradients to behave similar to PooledGD?
> >
> > Yes, you are right. If the batch size is not large enough, PooledSGD may not converge.
> >
> > >  I would request the authors to include numerical simulations for cases when some of the assumptions (eg. batch size) are violated,....
> >
> > Thank you for your kind suggestion. We will add more simulations on the effects of varying batch size in the camera-ready version.

---

### Official Review · Reviewer_uFbc · 2024-07-15

**Soundness:** 3
**Presentation:** 3
**Contribution:** 3
**Rating:** 5
**Confidence:** 3

**Summary:**

This paper studies the difference between the solutions to multi-environment matrix-sensing obtained by gradient descent and "heterogenous" stochastic gradient descent. The authors show through analytical results and simulations that HeteroSGD helps discover invariant solutions while gradient descent converges to solutions that contain both invariant and spurious components. The authors describe multi-environment matrix sensing as a mathematical model where the matrices being sensed consist of an invariant component and a spurious component. The solutions of two optimization problems are considered: a) gradient descent on a pooled batch of data from all environments and b) heteroSGD where each iteration samples a batch of data from a different environment. The authors introduce assumptions on the near-orthogonality between the invariant and spurious subspaces, the heterogeneity of the environments, and that the measurements satisfy RIP. Under these conditions, they show that heteroSGD discovers the invariant signal while gradient descent converges to solutions that contain both invariant and spurious components. The authors also run simulations to support their claims.

**Strengths:**

The paper is well written and easy to follow. All assumptions are stated clearly and the authors provide a baseline (gradient descent) to compare their proposed algorithm against.

**Weaknesses:**

1. The problem of multi-environment matrix sensing seems under-motivated. While the authors introduce a clean mathematical model for this problem, they do not provide instances of problems that satisfy this model. Is it reasonable to model environment specific signals as being incoherent with the invariant signal? Is it reasonable to assume heterogenous environments? In what situations are these assumptions satisfied? While I understand the focus of this paper is theoretical analysis of a model, I think it is important to motivate it so that we understand this is a real problem and not just another mathematical model.

2. The proof sketch section does not provide any intuition about the separation between GD and heteroSGD. Specifically, I'm looking for an explanation as to why $\mathbf{Q}_t$ stays small when environment specific data is provided (as in heteroSGD) and it does not decay when the environment data is pooled (as in GD). (If the answer is already in the paper I might have missed it - please point me towards it).

3. Simulations are run on synthetic examples. Are there any real datasets that you can demonstrate your methods on?

4. The current theorem statements show that under conditions of heterogeneity, heteroSGD can learn invariant solutions. Can one show that heterogeneity is necessary for learning invariant solutions?

**Questions:**

Some questions in weaknesses. Others below:

1. Did the authors try searching for different learning rates for GD vs SGD? Prior results on the interaction between batch size and learning rate indicate that the learning rate for GD and SGD should likely be tuned separately. Does the spurious solution still exist when this is taken into account?

2. Does the separation between pooled data vs environment specific data hold when SGD is considered (rather than GD)? Is the invariant learning captured by stochastic gradients or the fact that each update only uses data from one environment? Hopefully this can be answered theoretically and through simulations.

**Limitations:**

Yes

---

> ### Author Rebuttal · Authors · 2024-08-04
>
> We would like to thank the reviewers for the valuable feedback and insightful comments. We have carefully considered your comments and questions and have addressed them as below:
>
> > The problem of multi-environment matrix sensing seems under-motivated. Instances of problems and reasonability.
>
> A: We fully understand your concerns regarding our formulation. We will add more real scenarios to exemplify our abstract problem. Let us first explain two aspects.
>
> - About the formulation of multi-environment. In practice, the data are often collected from different environments. The spurious signal refers to the **endogenous spurious variables**, which inherit the dataset bias and so have non-zero associations that are unstable and thereby should be eliminated, such as the **intervened children of the response variable $Y$** in Structural Causal Model (SCM) [1]. While many existing works propose different methods to achieve invariance learning, the implicit regularization effect for achieving invariance learning is not well studied. This is also the starting point of our work. Additionally, besides eliminating the endogenous spurious variables, our model learns the sparsity, which also helps eliminate **exogenous spurious variables** which do not contribute to predicting the response variable $Y$.
>
> - About the matrix sensing problem. The matrix sensing problem is widely used in implicit bias contexts as a testbed for understanding the loss landscape and training dynamics of over-parameterized deep learning models since it behaves like neural networks with non-convexity and non-linearity. However, it can be solved efficiently under suitable conditions. We thus hope our work can provide insights into the implicit invariance learning abilities of deep learning models. The incoherence condition is common in matrix sensing or more broadly in high-dimensional feature selection problems.
>
> [1] Glymour, M., Pearl, J., & Jewell, N. P. (2016). Causal inference in statistics: A primer. John Wiley & Sons.
>
> > About the intuition about ``HeteroSGD``.
>
> A: Thanks for this question. For ``HeteroSGD``, as illustrated informally in **line 248**, the oscillation creates a contraction effect, which helps prevent the model from fitting the spurious signal. In contrast, for ``PooledGD``, there is no oscillation and the model is consistently driven towards the averaged signal and cannot distinguish between the invariance signal and the spurious signal. We will provide more intuitions.
>
> > About the simulations?
>
> A: Thanks for raising the question. There are some works that empirically achieve invariance learning from multi-environments (e.g. [2]). Our work attempts to theoretically reveal how models can learn invariance from the standard training procedure, and our simulations intends to verify our theoretical results, as the first work from this perspective. We will consider generalizing our theoretical findings to design more empirical methods for invariance learning in the future.
>
> [2] Simple data balancing achieves competitive worst-group-accuracy
>
> > Can one show that heterogeneity is necessary for learning invariant solutions?
>
> A: Yes, heterogeneity is essential for distinguishing between spurious and invariant components of the signal. Conceptually, the signal is said to be invariant as it does not change across the environments. If there is no heterogeneity, it means all the environments are the same or there is only one environment, so all the signal would be invariant. From the technical viewpoint, if our proposed heterogeneity conditions are severely violated, one can construct counterexamples where standard optimization algorithms fail to learn invariant solutions.
>
> > About trying searching for different learning rates for GD vs SGD, and the interaction between batch size and learning rate.
>
> A: Thanks for the insightful comment. For learning rate, our results point out the range of learning rate for ``HeteroSGD``. For ``PooledGD``, its failure is because the signal is averaged when calculating gradients. Therefore, tuning the learning rate will not prevent the model from learning the spurious solution.
>
> In our current results of SGD, we do not treat the batch size as a tunable parameter. The batch size is set to satisfy the RIP condition, which is a very common condition in the field of matrix sensing. Emprically ``PooledSGD`` is not stable when batch size is very small. Here $(d,r_1,r_2,m)=(30,5,5,20)$:
>
> |  Learning rate $\eta$ of ``PooledSGD``   | 0.001 | 0.005 | 0.01  | 0.05  | 0.1    |
> | ----- | ----- | ----- | ----- | ----- | ------ |
> | $\min_{t\le T}\left\|\mathbf{U}_t\mathbf{U}_t^\top - \mathbf{A}^* \right\|_F$ | 1.682 | 1.820 | 2.125 | 2.185 | Trajectory exceeds boundary. |
> We will add representative figures in final version.
>
> > Does the separation holds when ``PooledGD`` is replaced by ``PooledSGD``?
>
> A: Yes. We will add the following theorem in the revised version of our paper.
>
> Theorem 3.1 (Negative Result for ``PooledSGD``). Under the assumptions of Theorem 3, if we perform SGD ending with $T=\Theta(\log d)$,  where each sample of a batch is randomly chosen from an environment w.r.t. $D$, and the linear measurements are symmetric Gaussian, and the batch size is $d\cdot\mathrm{poly}(r_1,r_2,M_1, \log(d))$, then with probability over 0.99,
> $$
> \left\| \mathbf{U}_T\mathbf{U}_T^\top - \mathbf{U}^*{\mathbf{U}^*}^\top-\mathbf{V}^*{\mathbf{V}^*}^\top \right\|_F = o(1),
> $$
> during which for all $t=0,1,\ldots, T$:
> $$
> \left\| \mathbf{U}_t\mathbf{U}_t^\top - \mathbf{A}^*\right\|_F\gtrsim \sqrt
> {r_1\wedge r_2}.
> $$
> The intuition is that the heterogeneity across environment is reduced. Hence the gradients will be very close to the gradients in ``PooledGD``.
>
> ----
>
> We again thank you for dedicating your time and effort to reviewing our manuscript. Your valuable questions and insightful comments have significantly improved our work. We hope our responses have addressed all your concerns.

---

### Decision · Program_Chairs · 2024-09-25

**Decision:**

Accept (poster)

**Comment:**

This submission investigates the implicit bias of SGD in multi-environment matrix sensing. All reviewers rated this submission positively. They appreciated that the submission is well-written and contains solid theoretical analysis. The theoretical findings are interesting and would enhance our understanding of invariance learning. The reviewers made several suggestions to improve the submission (e.g., additional experiments when the assumptions are violated). But, based on the discussion, these suggestions do not seem to warrant an additional round of review to reach an acceptance recommendation.